# The in-tissue molecular architecture of β-amyloid pathology in the mammalian brain

Conny Leistner[1], Martin Wilkinson [2], Ailidh Burgess [1,3], Megan Lovatt [1], Stanley Goodbody[1], Yong Xu [2,4], Susan Deuchars[1], Sheena E. Radford [2] ✉, Neil A. Ranson [2] ✉ & René A. W. Frank [1] ✉

Amyloid plaques composed of Aβ fibrils are a hallmark of Alzheimer's disease (AD). However, the molecular architecture of amyloid plaques in the context of fresh mammalian brain tissue is unknown. Here, using cryogenic correlated light and electron tomography we report the in situ molecular architecture of Aβ fibrils in the $App^{NL-G-F}$ familial AD mouse model containing the Arctic mutation and an atomic model of ex vivo purified Arctic Aβ fibrils. We show that in-tissue Aβ fibrils are arranged in a lattice or parallel bundles, and are interdigitated by subcellular compartments, extracellular vesicles, extracellular droplets and extracellular multilamellar bodies. The Arctic Aβ fibril differs significantly from an earlier $App^{NL-F}$ fibril structure, indicating a striking effect of the Arctic mutation. These structural data also revealed an ensemble of additional fibrillar species, including thin protofilament-like rods and branched fibrils. Together, these results provide a structural model for the dense network architecture that characterises β-amyloid plaque pathology.

Alzheimer's disease (AD) results in cognitive decline and brain atrophy that is characterised by multiple pathologies, including the formation of abnormal extracellular protein deposits of β-amyloid (Aβ), intracellular tangles of tau, alongside neuroinflammation and the loss of neurons and synapses[1]. Mutations identified in familial forms of AD (FAD) indicate a causal role for the amyloid precursor protein (*APP*) and presenilin (*PSEN1/PSEN2*) genes, which encode the precursor of Aβ and the ɣ-secretase enzyme that catalyses the final step of Aβ peptide production, respectively[2,3]. ɣ-Secretase produces Aβ peptides that vary in length, of which $Aβ_{1-40}$ and $Aβ_{1-42}$ are most abundant. These peptides are highly aggregation prone, assembling into diffusible, low-molecular weight oligomers or protofibrils that precede the formation of larger Aβ fibrils[4]. Over decades, Aβ peptides, particularly $Aβ_{1-42}$[5], accumulate and form amyloid plaques in the parenchyma of the AD brain[2]. Amyloid plaques have been categorised as diffuse, dense-cored, fibrillar or neuritic, all of which contain fibrillar Aβ deposits[6].

Conventional EM of plastic embedded tissue from post-mortem AD brain[7,8] and FAD animal models[9] suggest that plaques are composed of parallel bundles and a lattice of Aβ fibrils. Additionally, conformation-specific fluorescent dyes suggest a heterogeneity of Aβ conformations within distinct regions of amyloid plaques[10,11].

Structures of amyloid fibrils of Aβ have been determined recently by cryoEM of fibrils assembled in vitro from recombinant or synthetic $Aβ_{1-42}$[12], as well ex vivo fibrils purified from post-mortem human AD brain and FAD mouse models[13,14]. These ex vivo samples yielded two structural arrangements for fibrils within $Aβ_{1-42}$ amyloid, with type I fibrils associated with sporadic disease and type II associated with familial disease and other brain pathologies[13]. Both forms differ from $Aβ_{1-42}$ fibrils prepared in vitro[12] and from $Aβ_{1-40}$ prepared from the meninges of cerebral amyloid angiopathy (CAA) cases[15]. However, the molecular architecture and organisation of β-amyloid within plaques from fresh, unfixed, brain tissue remained unknown.

[1]Astbury Centre for Structural Molecular Biology, School of Biomedical Sciences, Faculty of Biological Sciences, University of Leeds, Leeds LS2 9JT, UK. [2]Astbury Centre for Structural Molecular Biology, School of Molecular and Cellular Biology, Faculty of Biological Sciences, University of Leeds, Leeds LS2 9JT, UK. [3]Present address: Francis Crick Institute, 1 Midland Road, London NW1 1AT, UK. [4]Present address: AstraZeneca, 1 Francis Crick Avenue, Cambridge CB2 0AA, UK. ✉e-mail: s.e.radford@leeds.ac.uk; n.a.ranson@leeds.ac.uk; r.frank@leeds.ac.uk

Here we sought to visualise Aβ fibrillar plaques in situ using a homozygous, knockin FAD mouse model ($App^{NL-G-F/NL-G-F}$)[16] by preparing fresh, hydrated, vitrified tissue samples for cryogenic correlated light and electron microscopy (cryoCLEM) and cryo-electron tomography (cryoET). We also determined the ex vivo, 3.0 Å resolution structure of Aβ fibrils purified from the same $App^{NL-G-F}$ mouse brains by single particle cryoEM. Analysis of the fibrils within in-tissue tomograms revealed the presence of fully assembled fibrils, along with protofilament-like rods which may describe early assembly intermediates, and branched fibrils, suggestive of secondary nucleation mechanisms occurring in vivo. These data describe a dense network of fibrils, interdigitated with non-amyloid constituents that defines the in-tissue 3D molecular architecture of the amyloid plaque.

## Results

To determine the molecular architecture of pathology within fresh, intact tissue we developed a cryoCLEM workflow that we applied to 11-14 month-old $App^{NL-G-F}$ mice[16]. These animals have a humanised mouse *App* gene containing three familial Alzheimer's disease mutations (Swedish, Beyruthian, and Arctic)[16]. The Swedish and Beyruthian mutations are located upstream and downstream of the coding region for the Aβ peptide, and increase the overall Aβ concentration, and the ratio of Aβ$_{1-42}$:Aβ$_{1-40}$, respectively. In contrast, the Artic mutation is situated within the Aβ peptide coding sequence (App E693G, Aβ E22G) and is thought to increase the generation of Aβ protofibrils[17]. $App^{NL-G-F}$ mice develop β-amyloid plaques, neuroinflammation, damaged synapses, and behavioural phenotypes, without ectopic over-expression of APP[18]. To identify amyloid pathology within fresh tissue, mice received an intraperitoneal injection of a fluorescent amyloid dye, methoxy-X04 (MX04; Fig. 1a)[19]. Immunohistochemical fluorescence imaging confirmed MX04 detected β-amyloid in $App^{NL-G-F}$ mice (Supplementary Fig. 1a, b).

To cryopreserve anatomically intact tissue, 2 mm cortical biopsies of acute brain slices were high-pressure frozen and imaged by cryogenic fluorescence microscopy (cryoFM), identifying the locations of MX04-labelled amyloid plaques (Fig. 1a, b). We used these cryoFM maps of amyloid deposits to target the preparation of tissue cryo-sections by trimming a $60 \times 100 \times 150$ μm stub of tissue containing a single MX04-labelled amyloid plaque, from which 70-150 nm thick tissue cryo-sections were collected and attached to an EM grid support (Fig. 1a)[20–22]. CryoFM of $App^{NL-G-F}$ tissue cryo-sections revealed 5-20 μm star-shaped or round β-amyloid pathology (Fig. 1c). These were similar to plaques observed by fluorescence imaging of fixed samples and were absent in MX04-injected wild-type control samples (Fig. 1c and Supplementary Fig. 1b, c). To verify that $App^{NL-G-F}$ amyloid formed as extracellular deposits, we labelled tissue using an extracellular fluorescent marker, dextran-AF647 that is incapable of crossing the plasma membrane[23]. CryoFM of tissue cryo-sections from these samples indicated MX04 overlapping with dextran-AF647, confirming the extracellular location of amyloid (Fig. 1d). MX04-labelled amyloid plaques were mapped on medium magnification electron micrographs by cryoCLEM (Fig. 1c) to target areas for the collection of tomographic tilt series, which each encompassed a 1.3 μm² area of the tissue cryo-section. We collected 23 tomograms (4 with and 19 without a Volta phase plate) sampling central and peripheral regions of MX04-labelled plaques (Supplementary Data File 1).

### The native 3D molecular architecture of amyloid plaques

Reconstructing 3D cryotomographic volumes revealed the native, in-tissue cellular and molecular architecture of amyloid pathology (Fig. 2, Supplementary Movies 1 and 2 and see 'Methods' describing criteria for identifying macromolecular and cellular constituents within cryoET data). A salient feature of every tomogram that by cryoCLEM colocalised with MX04-labelled amyloid was the presence of dense arrays of fibrils. (Figs. 2a–c and 3 and Supplementary Data File 1). These

were arranged as a lattice or in parallel bundles (Supplementary Fig. 2), consistent with previous observations using conventional (fixed, plastic embedded, heavy metal-stained) EM[7–9, 24].

In central regions of MX04-labelled plaques, fibrils were interdigitated with extracellular vesicles (Fig. 2 and Supplementary Fig. 3). These were readily separable into different types (Supplementary Data File 1): (i) Extracellular vesicles (50–200 nm diameter), (ii) extracellular vesicles containing a cup or C-shaped membranes in their lumen, (iii) ellipsoidal vesicles (5–20 nm diameter). Multilamellar bodies composed of vesicles wrapped in multiple concentric rings of membrane were also apparent in 30% of tomograms (Figs. 2c and 4). Similar intracellular intermediates of autophagy have been detected by conventional EM of AD brain[25]. Extracellular droplets were also present in 30% of all in-tissue tomograms (Figs. 2 and 4). These were smooth, 80–120 nm diameter spheroidal structures that were similar, albeit fivefold to tenfold smaller, to the lipid droplets that reside intracellularly within healthy cells[26].

We next quantified whether extracellular vesicles were more common in MX04-labelled amyloid plaques than in non-AD tissue. We previously reported the in-tissue molecular architecture of mouse brain by cryoET of cryo-sections from $Psd95^{GFP}$ knockin mice that do not contain FAD mutations ($App^{WT/WT}$)[27]. This mouse strain labels glutamatergic synapses with GFP-tagged PSD95 but were otherwise without fibrils, amyloid plaques, or any other abnormal phenotype[27–29]. Comparing the prevalence of extracellular vesicles in $App^{NL-G-F}$ in-tissue tomograms with those from $App^{WT/WT}$ - $Psd95^{GFP/GFP}$ mice (Supplementary Data Files 1 and 2) indicated that there were on average 100-fold more extracellular vesicles in MX04-labelled amyloid plaques compared to that of tissues lacking β-amyloid plaque pathology ($P < 0.005$, mean=10.6 and 0.1, $n = 18$ and $n = 40$ in $App^{NL-G-F}$ and 40 App$^{WT/WT}$-$Psd95^{GFP}$ in-tissue tomograms, respectively, Fig. 3c and Supplementary Fig. 4). Interestingly, only exosomes were present in App$^{WT/WT}$-$Psd95^{GFP}$, whereas C-shaped membranes, ellipsoidal vesicles, multilamellar bodies, and extracellular droplets were absent in tissues lacking amyloid plaques (Supplementary Fig. 4).

At the periphery of MX04-stained plaques subcellular, membrane-bound compartments were observed interdigitated or wrapped around amyloid (Figs. 3a, b and 4, Supplementary Fig. 5, and Supplementary Movies 3 and 4). In these tomographic volumes the boundary between intra- and extracellular region of tissue was marked by plasma membranes that enclosed characteristic cytoplasmic constituents (see 'Methods'), including putative ribosomes, microtubules, as well as rough endoplasmic reticulum and mitochondria (Figs. 3 and 4 and Supplementary Figs. 5 and 6). In accordance with the colocalization of MX04 and the dextran-AF647 extracellular in tissue cryo-sections (Fig. 1d), there was no indication of β-amyloid fibrils within the cytoplasm of these cells. The observation of cells at the periphery of amyloid plaques is consistent with the microglia and astrocytes that typically surround extracellular amyloid deposits[8].

### The high-resolution structure of Arctic Aβ fibrils

We next sought to investigate how the in-tissue architecture of β-amyloid plaques can be reconciled with atomic models of Aβ fibrils. Recent structures of Aβ amyloid fibrils from $App^{NL-F}$ mice that express wild-type Aβ$_{1-42}$ form type II Aβ fibrils[13]. The structure of amyloid fibrils that result from the $App^{NL-G-F}$ knockin strain, that only differs from $App^{NL-F}$ in the Arctic familial AD mutation (E22G) within the Aβ$_{1-42}$ peptide, had not been reported. We therefore performed a sarkosyl-extraction of Aβ fibrils from $App^{NL-G-F}$ mouse forebrains, for single-particle cryoEM, using procedures developed by Yang and co-workers[13]. The sample contained fibrils with an overt helical twist corresponding to a predominant crossover distance of ~66 nm (Supplementary Fig. 7a), allowing us to determine a 3.0 Å resolution structure of the fibrils from $App^{NL-G-F}$ mouse brain (Fig. 5a, b and Supplementary Fig. 7). The fibril has a pseudo-$2_1$ screw axis, allowing us to

build an unambiguous atomic model of two identical copies of residues 1-38 of the $A\beta_{1-42}$ sequence into the cryoEM density map (Fig. 5c). We also observed a minor population of wider fibril segments composed of an apparent dimeric assembly of fibrils containing four copies of $A\beta$ per molecular layer that had too few particles to obtain a helical solution (Supplementary Fig. 7c, d). Consistent with earlier MS imaging[30], mass spectrometry of ex vivo purified amyloid identified mainly $A\beta_{1-42}$ and smaller peaks corresponding to $A\beta_{1-38/39}$ $A\beta_{11-42}$ and $A\beta_{1-40}$ suggesting that 61% of C-terminal residues 39-42 of $A\beta_{1-42}$ are present in the fibrils (Supplementary Fig. 8a, b), but are disordered in our map.

$App^{NL-G-F}$ ex vivo fibrils, which had a left-handed twist (as determined by cryoET, Supplementary Fig. 9 and 'Methods'), have a differ-ent structure than that found in fibrils extracted from $App^{NL-F}$ mice and AD brain, which lack the Arctic E22G mutation. The backbone con-formation of the $A\beta$ peptides were arranged as two 'S'-shaped proto-filaments that form an extensive inter-protofilament interface. The ordered core of each protofilament contains residues 1-38, compared to residues 9–42 in type I and II wild-type $A\beta_{1-42}$[13]. The solvent-accessible surface was also fundamentally different (Supplementary Fig. 8c). In both type I and II fibrils formed from wild-type $A\beta_{1-42}$, E22 is surface exposed[13]. However, in Arctic $A\beta_{1-42}$ fibrils, which only differ in the E22G mutation, G22 is buried within the fibril structure, at a point near the intra and inter-protofilament interfaces (Fig. 5d–f). Thus, it appears likely that wild-type $A\beta$ would be unable to adopt this Arctic $A\beta$ amyloid fold because of steric clashes created by a Glu at position

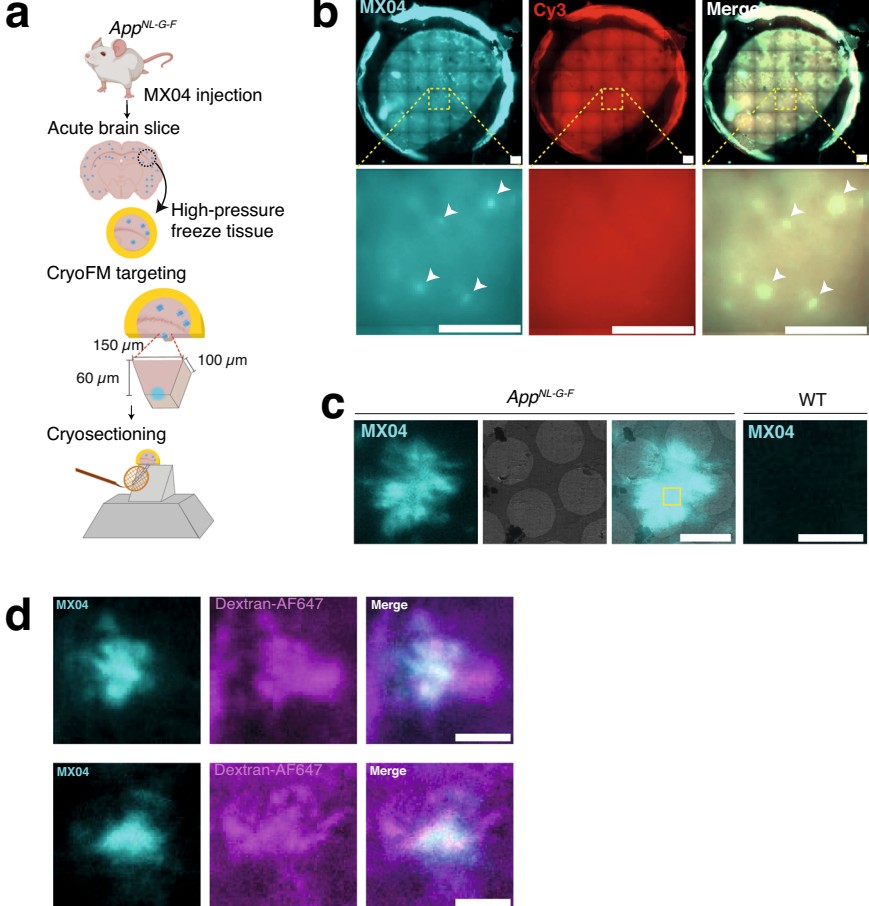

**Fig. 1 | Correlated light and electron microscopy to target the in-tissue pathology in the mammalian brain. a** Schematic showing correlative imaging work-flow to prepare fresh tissue for in-tissue cryo-electron tomography. $App^{NL-G-F}$ knockin mice were systemically administered with the amyloid fluorescent dye (methoxy-X04, MX04). Fresh acute brain slices were prepared, and tissue biopsies were high-pressure frozen. MX04-labelled plaques were detected by cryogenic fluorescence microscopy (cryoFM) within vitreous frozen tissue and used to target trimming of a trapezoid tissue stub to contain an amyloid plaque by cryo-ultramicrotomy. CEMOVIS/cryo-sections were collected from the front face of the amyloid plaque-containing stub and attached to the holy carbon support of an EM grid. Cryo-sections were 70–150 nm-thick and encompassed an area of ~130 × ~ 90 µm. **b** CryoFM imaging of a high pressure frozen $App^{NL-G-F}$ tissue biopsy. Top panels, image of whole tissue biopsy within circular gold carriers, and bottom panels, close-up. Left, Detection of MX04-labelled amyloid plaques (cyan puncta). Middle, control image of samples (excitation and emission of 550 and 620 nm, respectively) to detect non-specific autofluorescence and contamination. Right, merged image. Arrowheads indicate prominent amyloid plaques. Representative images of experiment that has been repeated independently >6 times with similar results. Scale bars, 200 µm. **c** Cryogenic correlated light and electron microscopy (cryo-CLEM) to locate amyloid within cryo-sections of $App^{NL-G-F}$ tissue and wild-type tissue control. First left, cryoFM detection of MX04-labelled amyloid plaque (cyan). Second left, cryoEM image of amyloid amyloid-containing tissue cryo-section. Second right panel, merged imaging showing aligned cryoFM and cryoEM image. Yellow box, region of amyloid plaque selected to collect the in-tissue tomogram shown in Fig. 2. First right, MX04-labelled wild-type tissue section serving as a control for the detection of amyloid by MX04. Representative images of experi-ment that has been repeated independently >6 times with similar results. Scale bars, 4 µm. **d** CryoFM of MX04-labelled $App^{NL-G-F}$ cryo-sections prepared from acute cortical slices were incubated with dextran-Alexa fluo-647 to label the extracellular space. Left, methoxy-X04 pseudo coloured cyan. Middle, dextran-Alexa fluo-647 pseudo-coloured magenta. Right, overlay of left and middle images. Scale bar, 20 µm. Representative images of experiment that has been repeated independently 3 times with similar results.

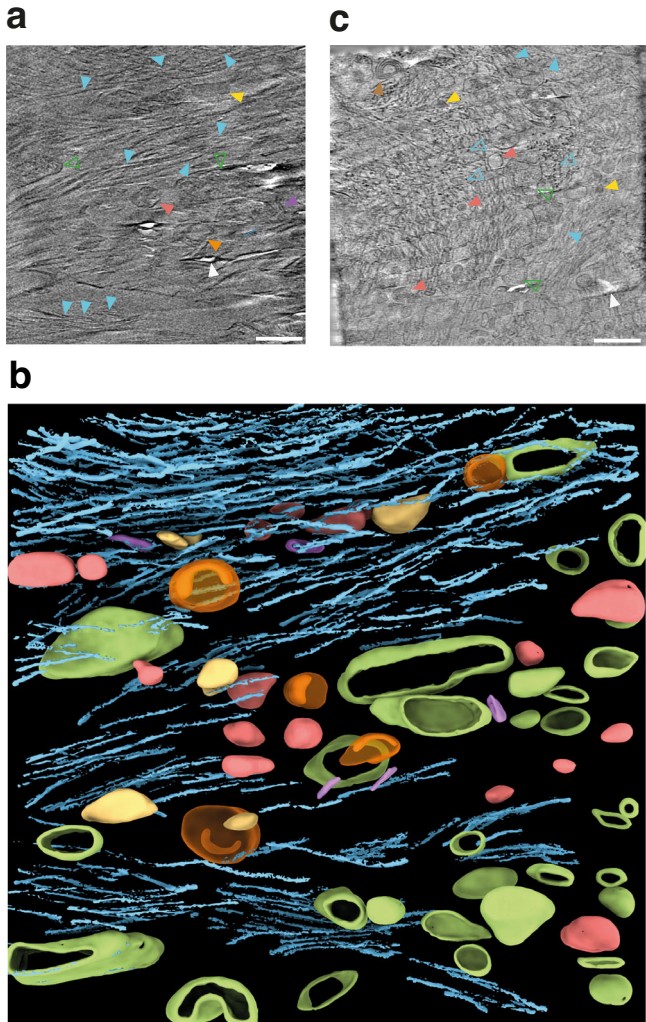

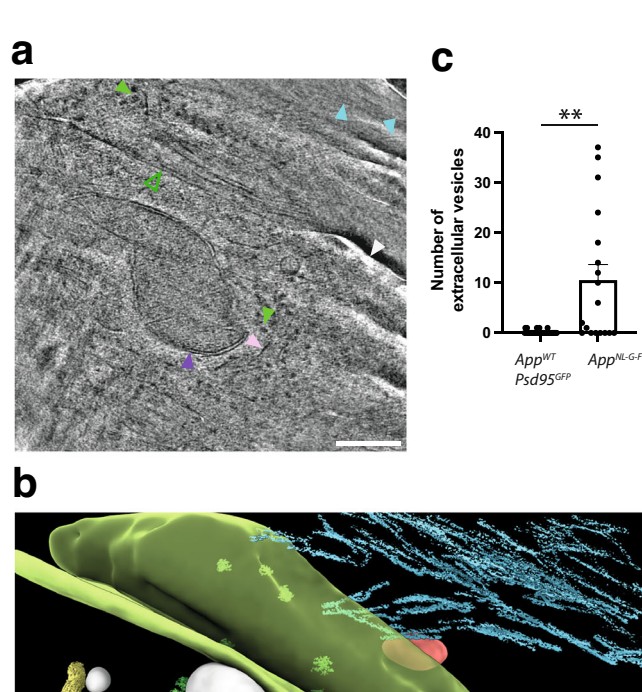

**Fig. 2 | CryoCLEM guided cryo-electron tomography showing pathology within central regions of MX04-labelled amyloid plaques. a** Tomographic slice through cryoET reconstruction of amyloid pathology within cortex showing cellular and molecular architecture. Filled cyan arrowheads, β-amyloid fibril oriented in the $x/y$ plane of the reconstructed tomogram. Open green arrowhead, plasma membrane of cellular compartment. These were larger than the volume of the tomogram and were therefore open at the top or bottom edge of the cryosection. Red arrowhead, spherical exosome. Orange arrowhead, C-shaped extracellular vesicle. Purple arrowhead, ellipsoidal extracellular vesicle. Yellow arrowhead, extracellular droplet. White arrowhead, localised knife damage in tissue cryo-section. Tomogram was collected with a Volta phase plate. Scale bar, 100 nm. **b** 3D segmentation of macromolecular and cellular constituents in a representative tomographic volume of a central region of an MX04-labelled amyloid plaque. Colours as in **a**. **c** Tomographic slice through cryoET reconstruction of pathology collected. Arrowheads as in **a**, except with the addition of open cyan arrowheads, β-amyloid fibril oriented in the along the $z$-axis of the reconstructed tomogram. Brown arrowhead, multilamellar body. Scale bar, 100 nm. Tomogram was collected without a Volta phase plate. See Supplementary Fig. 4 showing additional examples of tomographic data collected at central regions of amyloid plaques. See Supplementary Movies 1 and 2 showing example in-tissue tomographic volumes of encompassing central regions of amyloid plaques. See 'Methods' for criteria used to identify macromolecular and cellular constituents of tomograms.

**Fig. 3 | CryoCLEM guided cryo-electron tomography of amyloid plaques showing pathology at peripheral regions of MX04-labelled amyloid plaques. a** Tomographic slice through cryoET reconstruction of region at the periphery of MX04-labelled amyloid plaque. Cyan arrowhead, amyloid fibril. Open green arrowhead, plasma membrane. Closed light green arrowhead, ribosome. Purple arrowhead, mitochondria. Pink arrowhead, rough endoplasmic reticulum. Scale bar, 100 nm. See also Supplementary Fig. 5 showing additional examples of tomographic slices of peripheral regions of amyloid plaques. See Supplementary Movies 3 and 4 showing example in-tissue tomographic volumes of encompassing peripheral regions of amyloid plaques. **b** 3D segmentation of macromolecular and cellular constituents in a representative tomographic volume (region ii in Fig. 4) of peripheral region of an MX04-labelled amyloid plaque. Extracellular constituents: Cyan, amyloid fibril. Red, exosome. Cellular constituents: Green surface, plasma membrane. Green particles, ribosomes. Purple, mitochondria. Pink, rough endoplasmic reticulum. Yellow, microtubule. White, intracellular membrane vesicles. **c** The total number of extracellular vesicles per tomogram from $App^{NL-G-F}$ forebrain MX04-labelled amyloid plaques and $App^{WT/WT}$ - $Psd95^{GFP}$ forebrain lacking amyloid plaques plotted as mean ± SEM. **$p = 0.0048$, two-tailed Student's $t$ test, $n = 20$ tomograms from 2 $App^{NL-G-F}$ and 40 tomograms from 4 $App^{WT}$-$Psd95^{GFP}$ mice. See also Supplementary Fig. 4 and Supplementary Data set 1 and 2.

22 (Supplementary Fig. 8d, e). Alignment of available Aβ$_{1-42}$ structures from post-mortem human AD brains, and the two distinct Aβ amyloid structures from $App^{NL-F}$ and $App^{NL-G-F}$ mice, indicated that residues 20-36 form a very similar structure in each fibril (1.35 Å and 1.31 Å Cα RMSD A$pp^{NL-G-F}$ with Type I and Type II fibrils, respectively, Fig. 5g). This conserved structural motif adopts different positions with respect to

the fibril axis and the interface between symmetry-related protofilaments, giving rise to the different amyloid structures observed. The N- and C-terminal residues outside of this conserved region adopt different conformations, wrapping around this structural motif, or around the motif in its neighbouring, symmetry-related protofilament,

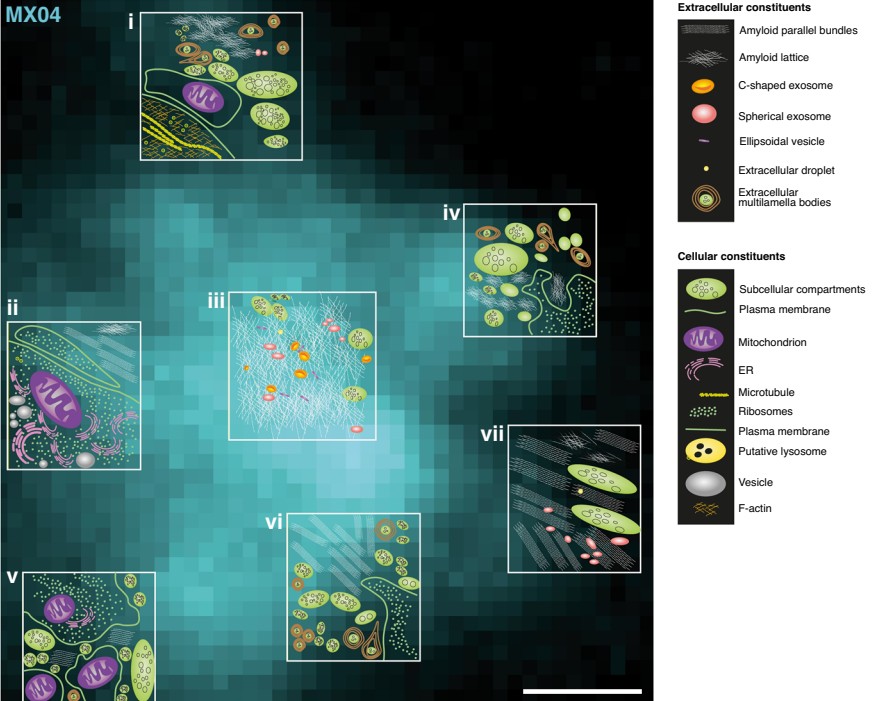

**Fig. 4 | A cryoET survey of the molecular architecture of a β-amyloid plaque.** Schematic representation of the native in-tissue 3D molecular architecture of an amyloid plaque overlaid on the MX04 signal of an amyloid plaque. Seven tomograms were collected (white boxes) at central and peripheral regions of a single amyloid plaque. The topology of molecular and cellular constituents is shown for each tomogram. An extracellular cryoCLEM marker (dextran AF-647) of this amyloid plaque is shown in Supplementary Fig. 6a. Tomographic slices corresponding to schematics ii and vii are shown in Fig. 3b and Supplementary Fig. 5a, respectively. Tomographic slices corresponding to schematics i, iii, iv, v, and vi are shown in Supplementary Fig. 6b-e, respectively. Scale bar, 1 μm.

or remaining unstructured, in these different amyloid folds. There does not appear to be any conservation between the ex vivo $A\beta_{1-42}$ structures and the fold of $A\beta_{1-40}$ amyloid fibrils (Fig. 5h) isolated from the meninges of human AD brains[15].

**Protofilaments and branched fibrils in amyloid plaques**

How well the structure of sarkosyl-extracted fibrils represents the fibrils observed in situ using cryoET was next addressed by collecting 27 tomograms of the sarkosyl extracted fibrils (Supplementary Fig. 9a and Supplementary Movie 5) and directly comparing the widths of fibrils in both ex vivo and in-tissue cryoET datasets (Fig. 6a). To rule out the possibility that MX04 altered the structure of the in-tissue fibrils, we also determined the structure of fibrils purified from MX04-injected $App^{NL-G-F}$ mice using single particle cryoEM, revealing a structure that was indistinguishable from unlabelled β-amyloid (Supplementary Fig. 10a, b). The width of lipid membrane bilayers served as an internal 'molecular ruler' of the accuracy of distance measurements within in-tissue tomograms, yielding an average thickness of $5.01 \pm 0.38$ nm (mean width $\pm$ SD; Fig. 6a), as expected[31]. The average width of ex vivo purified fibrils measured in tomographic slices ($9.2 \pm 2.8$ nm mean $\pm$ SD fibril width) (Fig. 6a) was consistent with the width of the reprojected 3D atomic structure of the fibrils determined using cryoEM ($8.5 \pm 0.95$ nm mean $\pm$ SD fibril width; Supplementary Fig. 10c). Unexpectedly, the average width of in-tissue fibrils was significantly smaller than that of ex vivo fibrils in tomographic volumes ($6.0 \pm 1.4$ nm versus $9.2 \pm 2.8$ nm mean $\pm$ SD fibril width, respectively; $p < 0.0001$). This difference could indicate a variation in the proportions of distinct fibril populations.

To assess further the possibility that distinct fibril populations were present in tissue, we tested whether fibrils of different width were evenly distributed across amyloid plaques by performing one-way analysis of variance, which indicated that there was a significant enrichment of thin versus thick fibrils in different in-tissue tomograms (Supplementary Fig. 11a; One-way ANOVA: $F(df=11) = 39.13$, $p = 2 \times 10^{-16}$). This suggested the variance of fibril width is region-specific. Importantly, all in-tissue tomograms showed an MX04 cryoCLEM signal in the location of regions enriched in either thin or thick fibrils, suggesting that both are composed of amyloid. However, fibril widths did not segregate on the basis of their location, i.e. whether they are at the periphery or in the core of amyloid plaques (Supplementary Fig. 11b).

Closer examination of the population of thick (7–11 nm width) fibrils in less crowded regions of in-tissue and ex vivo tomograms showed apparent crossovers (Fig. 6b, c). The helical pitch of these fibrils, corresponding to the distance between crossovers of ex vivo and in-tissue fibrils, was similar ($65.1 \pm 2.4$ nm mean $\pm$ SD versus $65.9 \pm 1.6$ mean $\pm$ SD helical pitch, respectively) and consistent with that observed in the single particle reconstruction (Fig. 6b; 65.5 nm helical pitch). In contrast, the thin (3–5 nm width) fibrils lacked the characteristic crossover of ex vivo Aβ atomic structures (Fig. 6d). Since the width of 3–5 nm fibrils was half that of the Aβ fibril structure, we describe this form of amyloid as 'protofilament-like rods'. Closer inspection of this minority of protofilament-like rods in ex vivo tomograms revealed that they were found branching away from a fibril (Fig. 6e and Supplementary 11c–f). Some branch points gave rise to protofilament-like rod extensions that were thinner (3-5 nm width) than their parent fibril (7–11 nm width), whereas other branches connected two 7–11 nm fibrils (Supplementary Fig. 11c).

To explore whether amyloid branching exists within in-tissue tomograms, we examined the raw tomographic density of parallel fibril bundles oriented along the tomographic z-axis (19 of 22 tomograms). These provided sufficient contrast to trace individual fibrils throughout the tomographic volume (Fig. 6f), revealing branched fibrils intermingled with unbranched fibrils within amyloid plaques (Fig. 6g, h and Supplementary Fig. 12a, b) in all 19 tomograms,

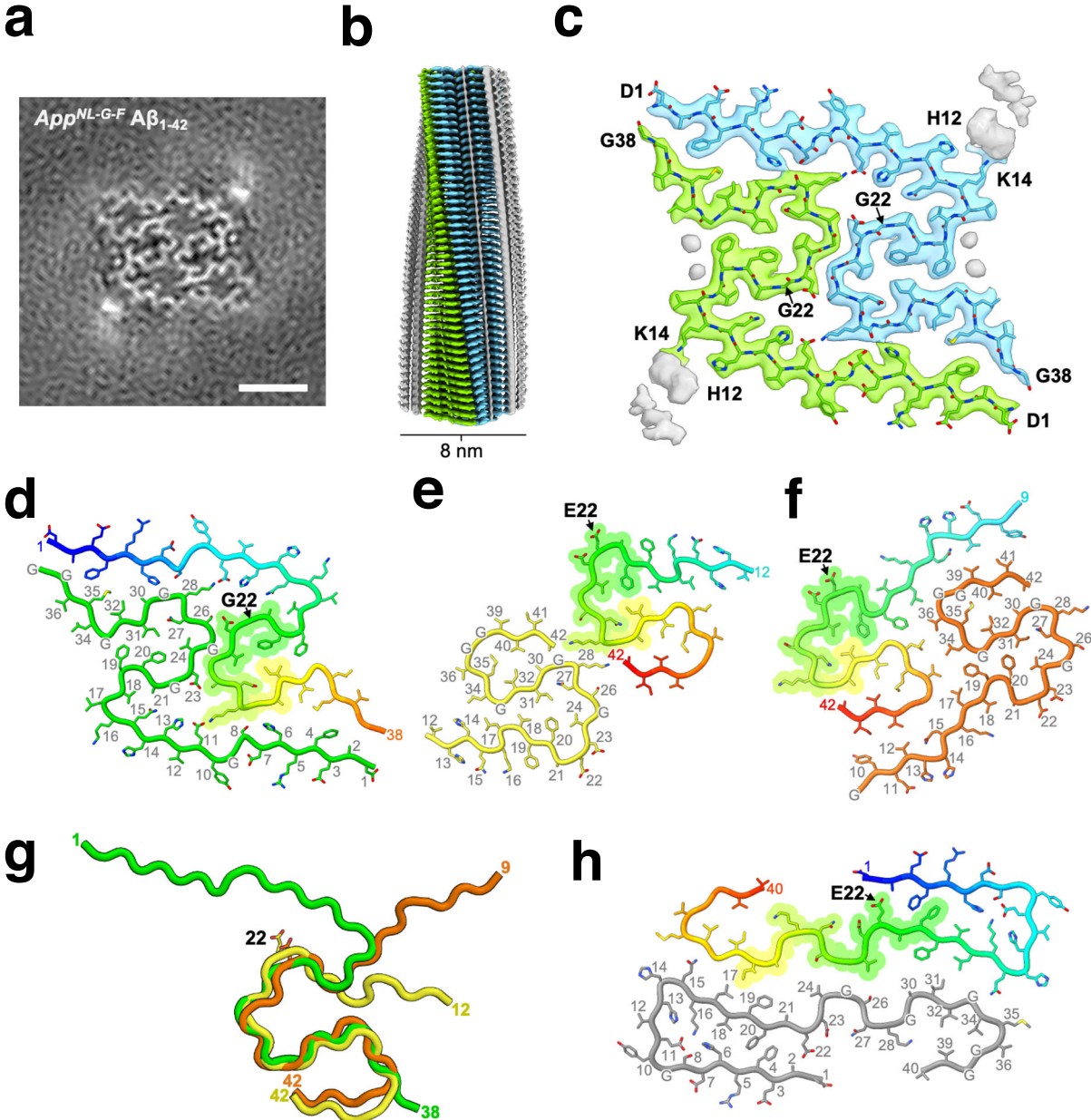

**Fig. 5 | The atomic model of $App^{NL-G-F}$ Aβ amyloid determined by single-particle cryoEM reconstructions from sarkosyl-extracts. a** Central section through the unmasked final $App^{NL-G-F}$ amyloid cryoEM map representing a single helical layer (~4.8 Å, 6x slices averaged) determined at a resolution of 3.0 Å (FSC 0.143). Scale bar, 2.5 nm. **b** Perpendicular view of the cryoEM map along the fibril axis, colour-coded by the protein chain (cyan and green), with unmodelled associated density in grey. The image shows clear resolution of individual β-strand layers within the amyloid core. **c** The cryoEM density and fitted model sectioned around a single helical layer, colour-coded as in **b**, with unmodelled associated density in grey. **d** One layer of the $App^{NL-G-F}$ β-amyloid model with one subunit coloured blue-to-red from N-to-C-terminus (the colouring includes the disordered residues 39–42 absent in the model). **e** Type II Aβ42 fibril model[13], with each subunit coloured as in d. Note that residues 1-38 are ordered in the Arctic structure, while residues 9–42 are structured in the type II fibrils. **f** Type I Aβ42 fibril model (PDB: 7Q4B)[13], coloured as in **d**. Note that residues 12-42 are structured in type I fibrils. **g** Superposition of the subunit folds from the different Aβ1-42 fibril models, showing only the aligned chain from each model with the chain colours correlating to those in d–f. This reveals a common protein conformation between residues 20–36, with diverging terminal folds. The per atom RMSD values for residues 20–36 between the models is: 1.35 Å for AppNL-G-F vs Type I and 1.31 Å for AppNL-G-F vs Type II fibrils. **h** Extracted Aβ40 fibril model from the meninges of Alzheimer's patient brains (PDB: 6SHS)[15], coloured as in **d**. Note that all residues 1–40 are ordered in this fibril type. The Aβ40 model does not show conservation of structural elements in relation to the other fibril folds displayed.

regardless of their location at the periphery or in the core of a plaque. The atomic structure of Arctic Aβ was well accommodated in the raw tomographic map of unbranched fibrils (Fig. 6h and Supplementary Fig. 12c), albeit fibrils in situ showed variable degrees of curvature, whereas the ex vivo single particle Arctic Aβ cryoEM model has helical symmetry imposed and is thus perfectly straight. Thus, in-tissue tomographic data suggest that in addition to Aβ fibrils containing two,

intertwined protofilaments with the Aβ Arctic amyloid fold, amyloid plaques also contain protofilament-like rods and branched amyloid structures.

## Discussion

Tomograms of Aβ amyloid in-tissue describe a complex molecular architecture of β-amyloid plaques that includes fibrils, protofilament-

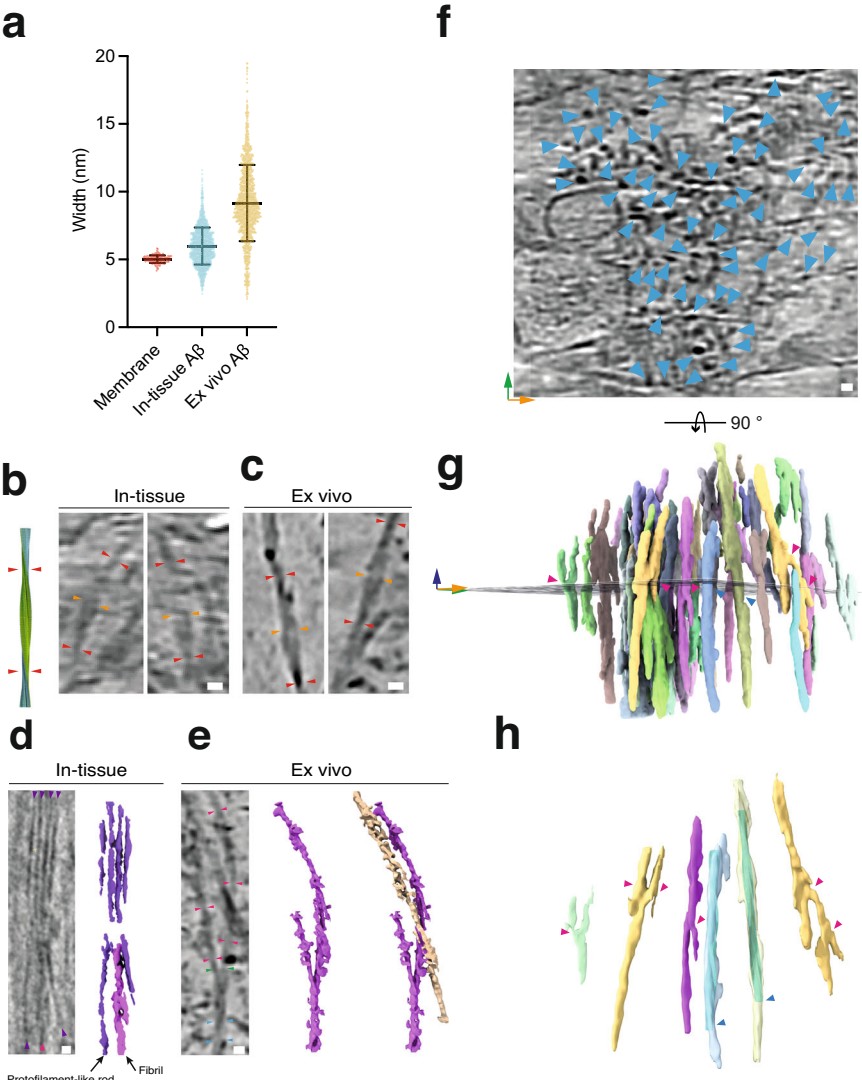

**Fig. 6 | A diversity of fibril types within amyloid plaques. a** Scatterplot showing the distribution of lipid membrane (*n* = 541) and fibril width from in-tissue (cryoET volumes of cryo-sections from *App^NL-G-F* amyloid plaques, *n* = 1840) and ex vivo (cryoET of sarkosyl-extracted amyloid fibrils, *n* = 1470). Middle and top/bottom clack bars indicate mean and one standard deviation. Raw data are provided in the Source data file. **b** Left, atomic model Arctic Aβ fibril. Right, tomographic slices showing in-tissue fibrils with an apparent helical twist of two protofilaments. Red arrowheads, apparent crossover. Orange arrowheads, point of maximum fibril width. Scale bar, 10 nm. **c** Same as **b** but for ex vivo amyloid. Scale bar, 10 nm. **d** Left, tomographic slice, and right, tomographic density showing in-tissue amyloid fibrils with 3–4 nm width fibrils (purple arrowheads) and a 7–10 nm fibril (magenta arrowhead). Scale bar, 10 nm. **e** Left, tomographic slice of branched fibrils in ex vivo amyloid. Blue arrowheads, 7-10 nm fibril. Green arrowheads, fibril branch point. Magenta arrowheads, protofilament. Scale bar, 10 nm. Middle, tomographic density of branched fibril. Right, Same as Middle except with nearby 7–13 nm fibril

(gold coloured) included. See Supplementary Fig. 11c–f showing additional examples of branched fibrils in ex vivo amyloid. See also Supplementary Movie 5 showing tomographic volume of ex vivo fibrils. **f** Tomographic slice of in-tissue amyloid with fibrils oriented on the *z*-axis of reconstructed tomographic volume. Cyan arrowhead, high-contrast spot corresponding to a single fibril. Orange and green arrows, *x*- and *y*-axis of tomogram, respectively. Scale bar, 10 nm. **g** Segmented tomographic density of fibrils shown in **c**, viewed from the side. Fibrils are coloured by connectivity. Magenta arrowhead, putative fibril branch point. Blue arrowhead, unbranched fibril. Orange, green, and blue arrows, *x*-, *y*-, *z*-axis, respectively. **h** Raw tomographic density of a subset of fibrils from **g**. Magenta arrowhead, putative branch point. Blue arrowhead, atomic model of Arctic Aβ fitted into the tomographic density of unbranched fibril (see 'Methods'). See Supplementary Fig. 12 showing an additional example of in-tissue branched amyloid. See Supplementary Movie 2 showing parallel bundles of fibrils oriented on the *z*-axis of the in-tissue tomographic volume.

like rods and branched amyloid. This diversity of amyloid architecture occurs alongside non-amyloid constituents, including extracellular vesicles, extracellular multilamellar bodies and cellular compartments. These observations are consistent with earlier studies that used conventional EM of resin embedded post-mortem AD tissues[7,8], animal models[9], and tomography of resin-embedded in vitro cell-based model of Aβ fibril toxicity[24]. The detail afforded by cryo-preservation of native tissue and cryoET suggests the existence of distinct types of extracellular vesicles, including those containing a luminal C-shaped membrane. Extracellular droplets that resemble lipid droplets were

also observed. These had eluded identification in earlier studies[7–9,24]. These non-amyloid extracellular constituents were either absent or rarely observed in normal brain tissue[22,27], suggesting that they are indicative of pathology, perhaps related to earlier stages of amyloid plaque biogenesis[24,32], or an on-going cellular response to the accumulation of amyloid[33].

Using cryoEM we determined the native in-tissue 3D molecular architecture of extracellular amyloid plaques of the Arctic variant of Aβ in the mammalian brain, and atomic models of ex vivo amyloid purified from these tissues. The atomic model is similar to a recent

report of the same amyloid[34] and showed that $App^{NL-G-F}$ fibrils of the Arctic variant have a different fold to that observed in the $App^{NL-F}$ knockin model, indicating that the Arctic FAD mutation (E22G) causes a dramatic change in the fibril structure. The structural differences include the identity of the residues involved in the amyloid core and in the arrangement of the two protofilaments, which nonetheless, each contain a common structural element involving residues 20–36 of the 42-residue sequence. These structural differences, exposing different residues on the surface of Arctic Aβ fibrils (Supplementary Fig. 8c), could explain the inability of the diagnostic reagent Pittsburgh B to detect Arctic amyloid in PET imaging of patients[35] and animal models[36].

The single-particle reconstruction of Arctic Aβ fibrils describes the structure of the major species of amyloid in $App^{NL-G-F}$ mice. However, cryoET data collected from ex vivo, purified amyloid also revealed the existence of rare protofilaments and branched fibrils. The presence of protofilament-like rods and fibril branch points was also apparent within in-tissue tomograms of $App^{NL-G-F}$ amyloid plaques. Protofilaments and branched fibrils have also been observed previously by atomic force microscopy of in vitro Aβ$_{1-42}$ assembly assays[37], albeit not all reported in vitro preparations of Aβ have shown this architecture[38]. Short protofilaments branching from fibrils have been observed by cryoEM imaging of in vitro Aβ aggregation assays[39]. The existence of protofilament-like rods and branched fibrils in situ provides a rationale for the dense architecture of plaque cores. Such branch points could result from catalysis of fibril growth in vivo by secondary nucleation[39,40] and provide a structural model for the focal accumulation of Aβ within amyloid plaques.

The structural diversity of Aβ is recognised as one of the challenges in developing therapeutics to treat Alzheimer's disease. Importantly, antibodies developed as an immunotherapy and tested in recent successful clinical trials were raised against the Arctic Aβ variant[41], which on the basis of in vitro prepared Aβ assemblies was expected to generate a high concentration of small, worm-like protofibrils[17]. Our in-tissue tomographic data of Arctic Aβ from murine FAD models reveal no such enrichment of short protofibrils. However, it is possible that other in situ structures of Aβ, including protofilament-like rods and branched amyloid contain equivalent epitopes to in vitro prepared protofibrils that mediate the effect of Arctic Aβ-directed immunotherapies. Future studies are needed to explore further the origin and detailed structural differences of these amyloids and to understand the in-tissue molecular architectures of post-mortem AD brain.

## Methods

### Laboratory animals
Animals were treated in accordance with the UK Animal Scientific Procedures Act (1986) and NIH guidelines. Animal experiments were approved by the University of Leeds Animal Welfare and Ethics Committee. Two 11–12-month-old male $App^{NL-G-F}$ knockin mice[16] were used for in-tissue structural analysis by cryoCLEM and cryoET. Three male 11-13 month old $App^{NL-G-F}$ and three male wild-type controls we used for immunohistochemistry. Two male 11-13 month old $App^{NL-G-F}$ knockin mice were used for each ex vivo amyloid purification. Four male Psd95$^{GFP/GFP}$ 7-11 month old mice were used for the quantification of extracellular non-amyloid constituents.

### Preparation of acute brain slices
Mice received an i.p. injection of 5 mg/kg Methoxy-X04 (Tocris) in 10% w/v DMSO containing phosphate-buffered saline. 24 hours after injection, mice were injected with pentobarbital and intracardially perfused with N-methyl-D-glucamine (NMDG)-HEPES solution (93 mM NMDG, 2.5 mM potassium Chloride, 1.2 mM sodium hydrogen carbonate, 20 mM HEPES, 25 mM glucose, 5 mM sodium ascorbate, 2 mM thiourea, 3 mM sodium pyruvate, 10 mM magnesium sulphate

heptahydrate, 0.5 mM calcium chloride dihydrate, osmolality = 300–315 mOsmol/kg, pH 7.4)[42]. Brains were retrieved and 100 μm thick acute slices were prepared using a vibratome (speed 0.26 mm/s, Leica, VT1200S, Campden Instruments Limited blades, J52/11SS blades) in ice-cold carboxygenated NMDG-HEPES solution.

### Immunohistochemistry and confocal fluorescence microscopy
Free-floating acute brain slices were fixed in 4% paraformaldehyde (PFA), blocked in 5% BSA, 0.1% w/v Triton-X100 containing Tris Buffered Saline (TBS, 50 mM Tris-Cl, 150 mM NaCl) and after three washes with TBS, incubated with 6E10 anti-amyloid-beta 1-16 mouse IgG1 (1:750, Biolegend, 803001) in 0.1% w/v Triton-X100 containing TBS at 4 °C for 24 h. After three washes in TBS, slices were incubated with anti-mouse-IgG1-AF-633 (1:1000, Life Technologies, A21126) in 0.1% w/v Triton-X100 in TBS for 48 h at 4 °C. After 3 washes, slices were mounted in Vectashield (Vector Laboratories) on superfrost slides (Erpredia, J1810AMNZ) with coverslips (Academy, 0400-8-18). Images were captured with a confocal laser scanning microscope (Zeiss LSM 700) using a ×10/0.3 and a ×20/0.5 numerical aperture (NA) air objective lens, with frame size 1024×1024 pixels and 512×512 pixels, respectively (Methoxy-X04 excitation: 405 nm, emission; 435 nm; AF-633 excitation: 639 nm, emission; 669 nm).

### High pressure freezing
Acute brain slices were sampled by collecting 2 mm diameter cortical tissue biopsies. These were incubated in cryoprotectant (20% w/v 40,000 Dextran[22] in NMDG-HEPES solution) for ~45 min at RT. 100 μm deep wells of the specimen carrier type A (Leica, 16770152) were filled with cryoprotectant, tissue biopsies were carefully placed inside and covered with the flat side of the lipid-coated specimen carrier type B (Leica, 16770153) and high-pressure frozen (-2000 bar, −188 °C) using a Leica EM ICE.

### Cryo-ultramicrotomy
High pressure frozen sample carriers were imaged with a cryo-fluorescence microscope (cryo-FM, Leica EM Thunder with HC PL APO 50/0.9 NA cryo-objective) at −180 °C to determine the location of Aβ plaques. The brightest fluorescent signal with an excitation of 350/50 nm and emission of 460/50 nm underneath the surface of ice were chosen for cryo-sectioning. The distance from Aβ plaque of interest to the edge of the carrier was measured to target the collection of cryo-sections containing an amyloid plaque. Next, the carriers were transferred to a cryo-ultramicrotome (Leica EM FC7, −150°C) equipped with trimming (Trim 45, T1865 and trim 20, T399) and CEMOVIS (Diatome, cryo immuno, MT12859) diamond knives. A 100 × 100 × 60 μm trapezoid stub of tissue was trimmed that contained the target amyloid plaque, from which 70–150 nm thin sections were cut at −150 °C with a diamond knife (Diatome, cryo immuno, MT12859). Sections were picked up with a gold eyelash and adhered onto a glow discharged (Cressington glow discharger, 60 s, 10−4 mbar, 15 mA) 1.2/1.3 or 3.5/1, 300 mesh Cu grid (Quantifoil Micro Tools) using a Crion electrostatic gun[43] and micromanipulators[44].

### Cryogenic fluorescence microscopy
High pressure frozen tissue inside gold carriers and tissue cryo-sections were screened for fluorescence using a cryogenic fluorescence microscope Leica EM Thunder with a HC PL APO ×50/0.9 NA cryo-objective, Orca Flash 4.0 V2 sCMOS camera (Hamamatsu Photonics) and a Solar Light Engine (Lumencor). A DAPI filter set (excitation 365/50, dichroic 400, emission 460/50) was used to detect methoxy-X04 labelled amyloid. A rhodamine filter set (excitation 546/10, dichroic 560, emission 525/50), was used as a control imaging channel. The images were acquired with a frame size of 2048×2048 pixels. Tile scans of high-pressure frozen carriers were acquired with 17% laser intensity for 0.1 s. Z-stacks of ultrathin cryo-sections were acquired

with 30% intensity and an exposure time of 0.2 s. Images were processed using Fiji ImageJ[45].

## Cryogenic correlated light and electron microscopy (cryo-CLEM)

The location of amyloid plaques in ultrathin cryo-sections was assessed by cryogenic fluorescence microscopy based on Methoxy-X04 fluorescence (excitation 370 nm, emission 460–500 nm). Grid squares that contained a signal for Methoxy-X04 were selected for electron tomography. The alignment between cryoFM images and electron micrographs was performed using a Matlab script[46,47], in which the centres of 10 holes in the carbon foil surrounding the region of interest were used as fiducial markers to align the cryoFM and cryoEM images.

## Cryo-electron tomography imaging and reconstruction

Electron microscopy was performed with a ThermoFisher 300 keV Titan Krios G2, X-FEG equipped with Gatan K2 XP summit direct electron detector and BioQuantum energy filter, in the Astbury Biostructure Laboratory (ABSL) at the University of Leeds. Tomographic tilt series were collected from +60° to −60° in 2° increments using a dose symmetric tilt scheme[48] in serialEM[49]. Two of the datasets were collected with a Volta phase plate[50] conditioned to 0.4–0.7 Π rad with a 0.8 to 1.3 μm defocus. A further two datasets were collected with 70 μm objective aperture and 5.0 to 6.5 μm defocus. Each tilt increment received 2 s exposure (fractionated into 8 movie frames) at a 0.5 e⁻.Å².s⁻¹ dose rate, resulting in a total dose per tilt series of ~61 electrons and pixel size of 3.42 Å. Dose fractions were aligned and tomograms were reconstructed using patch tracking in IMOD[51]. Segmentation was performed using the coordinates of fibrils manually picked in IMOD and of the lipid bilayer of membranes in Dynamo[52]. Figures were prepared in ChimeraX using SIRT reconstructed tomograms that were deconvoluted with Isonet[53] using defocus values estimated with Gctf[54]. Part of the schematic in Fig. 1a was made with Biorender.com.

## Annotation and analysis of macromolecular constituents in tomograms

Annotation described in Supplementary Data File 1 was performed blind by two curators to catalogue the constituents of in-tissue tomograms. First, each curator annotated all SIRT reconstructed tomograms independently. Next, a third curator inspected and certified each annotation (Supplementary Data File 1). The following constituents were identified: (1) Amyloid fibrils were assigned on the basis of methoxy-X04 cryoCLEM labelling and rod-shape within extracellular locations of the tissue. (2) Extracellular vesicles were defined as membranes that were closed and situated with extracellular locations. Extracellular vesicles were subdivided into three structural categories. (2a) Spherical exosomes (50–200 nm diameter) enclosed by a single plasma membrane. (2b) C-shaped exosomes composed of cup-shaped membrane within the lumen of a spherical exosome. (2c) Ellipsoidal vesicles (5–20 nm diameter). (3) Extracellular droplets: 80–120 nm amorphous and smooth spheroidal particles that resembled lipid droplets. (4) Extracellular multilamellar bodies: contained 60–200 nm vesicle or subcellular compartment rapped in a spiral of membrane lipid bilayer. (5) Mitochondria: defined by the double membrane including outer membrane and inner mitochondrial cristae. (6) Putative ribosomes were distinguishable as ~30 nm particles with higher tomographic density than surrounding macromolecules, in accordance with their high nucleic acid content. (7) Rough endoplasmic reticulum: intracellular tubular membranes with ribosomes associated on the outside edge. (8) Microtubules: were identified as ~25 nm wide filaments. (9) Subcellular compartments were defined by containing a higher tomographic density than the extracellular space that is consistent with the higher concentration of proteins in the cell cytoplasm compared to the extracellular space. (10) Cell plasma membranes: marked the boundary between the higher tomographic density of the

cytoplasm and the lower tomographic density of extracellular interstitial space, as well as enclosing putative ribosomes, microtubules, mitochondria, and rough endoplasmic reticulum. (11) Knife damage: Tissue cryo-sections contained small regions in which the sample has been compressed, leaving a crevasse in the tissue that were readily identified as holes within the tissue[55].

The extracellular versus intracellular topology of locations within cryoET data was established on the basis of macromolecular and organelle constituents that was consistent with the expected cell cytoplasm, including microtubules, putative ribosomes, rough endoplasmic reticulum, and mitochondria. Cell cytoplasm also appeared to contain a higher concentration of globular proteins than the interstitial space. We verified our assignment of extracellular amyloid using an extracellular fluorescent (dextran-AF647) cryoCLEM marker for a one of the four datasets (Fig. 3c and Supplementary Data File 1).

The workflow for collecting in-tissue tomograms from $App^{WT/WT}$-$Psd95^{GFP/GFP}$ mice forebrain (cortex and hippocampus) cryo-sections was as previously described[27] and was the same as that described for $App^{NL-G-F}$ mice, but without i.p. administration of MX04. To quantify extracellular vesicles in tomograms from $App^{WT/WT}$-$Psd95^{GFP/GFP}$ mice (Supplementary Data File 2), the number of exosomes, extracellular vesicles, ellipsoidal vesicles, multilamellar bodies and extracellular droplets were counted and compared to those from $App^{NL-G-F}$ mice.

## Ex vivo purification and single-particle cryoEM data collection of amyloid from $App^{NL-G-F}$ mouse

Two 11–14-month-old homozygous $App^{NL-G-F}$ knockin mouse forebrains (cortex and hippocampus), with and without methoxy-X04 pre-treatment were used for each preparation of ex vivo amyloid following the sarkosyl extraction protocol reported by Yang and co-workers[13]. Briefly, two forebrains (hippocampus and cortex) were homogenised in 20 vol (w/v) homogenisation buffer (10% w/v sucrose, 1 mM EGTA, 800 mM NaCl, 10 mM Tris.Cl pH7.4) at 37 °C. The sample was then brought to 2% w/v sarkosyl and incubated at 37 °C for 60 min. All subsequent steps were performed at room temperature. The total extract was centrifuged at 10,000 × g for 10 min. The supernatant was retained and centrifuged at 100,000 × g for 60 min. The pellet was resuspended in 1 ml/g extraction buffer (2% sarkosyl in homogenisation buffer) and centrifuged at 3000 × g for 5 min. The supernatant was diluted threefold in wash buffer (0.2% w/v sarkosyl, 10% w/v sucrose, 150 mM NaCl, 50 mM Tris.Cl pH7.4) and centrifuged at 100,000 × g for 30 min. The pellet was resuspended in 300 μl/g EM buffer (20 mM Tris.Cl pH 7.4, 50 mM NaCl) and stored at 4 °C. Plunge frozen samples for cryoEM were prepared by applying 4 μl of sample to 60 s plasma cleaned (Tergeo, Pie Scientific) Quantifoil R1.2/1.3 grids. Grids were blotted and plunge-frozen in liquid ethane using a Vitrobot Mark IV (ThermoFisher) with a 1 s wait and 5 s blot time respectively. The Vitrobot chamber was maintained at close to 100% humidity and 6˚C. The single-particle datasets were collected at ABSL (Leeds) using a Titan Krios electron microscope (ThermoFisher) operated at 300 ke⁻V with a Falcon4 detector in counting mode (Nominal magnification of 96,000x and 0.83 Å/pixel). A total of 2,428 movies were collected using EPU-3.0 (ThermoFisher) with a nominal defocus range of −1.6 to −3.1 μm and a total dose of ~52 e⁻/Å² over an exposure of 8 s, corresponded to a dose rate of ~4.5 e⁻/pixel/s. The control methoxy-X04 dataset was collected with an additional Selectris energy filter with a 10 e⁻V slit, a nominal magnification of ×130,000 and a pixel size of 0.94 Å. A total of 4165 movies were collected with a nominal defocus range of −1.4 to −2.9 μm and a total dose of ~41 e⁻/Å² over an exposure of 6 s, corresponded to a dose rate of ~6.1 e⁻/pixel/s.

## Single particle helical reconstruction of $App^{NL-G-F}$ ex vivo Aβ fibrils

Raw EER movies were compressed and converted to TIFF using RELION-4[56], regrouping frames into 40 fractions to give a dose per frame of 1.3 e⁻/Å² for the $App^{NL-G-F}$ dataset and 1.0 e⁻/Å² for the methoxy-X04-treated

dataset. The TIFF stacks were aligned and summed using motion correction in RELION-4 and CTF parameters were estimated for each micrograph using CTFFIND4[57]. Fibrils from roughly 100 micrographs were picked manually and used to train a separate picking model for each dataset in crYOLO[58] for automated picking with an inter-box spacing of 3× helical repeats (~14 Å). Micrographs not containing fibrils were removed by screening the lowpass-filtered micrographs generated by crYOLO. This left 380 fibril-containing micrographs (Supplementary Fig. 7a) for further processing, from which 63,680 fibril segments were extracted 3× binned with 750 Å box dimensions. Two rounds of 2D classification were performed to remove picking artefacts, generating a cleaned dataset of 46,215 segments of fibrillar classes. The resulting class averages were split into two subsets based on fibril morphology, with 97% selected in the major subset and 3% showing wider fibrils in the minor subset (Supplementary Fig. 7b, c). Each subset was re-extracted 2× binned (500 Å box dimensions) and initial models were generated from a respective 2D class average with a helical twist estimate obtained from measured crossover distances using the relion_helix_inimodel2d command[59]. These templates were used to start a one-class 3D classification (initial lowpass filter of 15 Å) to obtain a refined initial model for each form (Supplementary Fig. 7b, c). A final 2D classification run on each 2x binned subset was used to remove unfeatured fibril segments. The structure of the wider, minor fibril form could not be accurately determined due to the low starting number of particles (Supplementary Fig. 7d). The cleaned 30,070 segments for the major subset were re-extracted unbinned (250 Å box dimensions) for further 3D classification runs with searches of the helical twist (Supplementary Fig. 7e). Only one ordered fibril form with a pseudo-$2_1$ helical symmetry was identified in the data from several classification attempts. After multiple rounds with increasing angular sampling and initial resolution of lowpass-filter, 2568 ordered segments were selected and the helical parameters optimised to give good layer separation (Supplementary Fig. 7f). The segments refined to 3.1 Å (gold-standard, 0.143 FSC) and then to a final resolution of 3.0 Å after CTF refinement and Bayesian polishing (Supplementary Fig. 7g, h). The final map was sharpened using a B-factor of −23 Å² and helical symmetry was applied using the refined helical twist of 179.352° and rise of 2.418 Å. The control, methoxy-X04 treated fibril dataset was similarly processed and yielded an identical structure at a resolution of 3.2 Å, deposited with a sharpening B-factor of −25 Å² (Supplementary Fig. 7a). Full details are shown in Supplementary Table 1.

Initially, we tried to dock the human Aß42 AD type I fibril structure (PDB: 7Q4B) into the map, which has a superficially similar structure. However, neither the overall model nor the register of the amino acid side chains fitted. We therefore built a de novo model for one chain using Coot[60] using the large bulky side chain densities, as well as the regions without side chain density (glycine residues) and general chemistry of the local environment to guide model building. Ramachandran and rotamer outliers were monitored and fixed as they appeared and then the chain was duplicated and rotated to fit a second interacting subunit density. The model was real-space refined using Phenix[61] and repeated to create a model for three helical layers containing six peptide chains in total before a final real-space refinement in Phenix with NCS and template restraints applied to limit chain divergence. The hand of the App$^{NL-G-F}$ ex vivo purified amyloid fibril twist was determined by cryoET (Supplementary Fig. 9). The final model quality was assessed using MolProbity[62] and the results detailed **in** Supplementary Table 1. The model was fit into tomographic density (Fig. 6h and Supplementary Fig. 12c) using ChimeraX[63,64].

### Mass spectrometry
Sarkosyl-extracted fibrils were resuspended in 500 µl hexafluoroisopropanol (Sigma-Aldrich) and sonicated in an ultrasonic water bath (Untrawave Ltd. Cardiff) for 5 min. The mixture was incubated at room temperature for 24 h and then centrifuged at 17,000 × g. for 5 min. The supernatant was filtered through a 0.22 µm PDVF filter (Millex-GV, Merck) and the filtrate was dried over a gentle stream of nitrogen gas to form a film of peptide around the wall of the tubes. The sample was freeze-dried for 24 hours. The mixture was then resuspended in 50% acetonitrile aqueous solution with 2% formic acid and centrifuged at 17,000 × g. for 5 min. The sample was subsequently analysed by ESI-MS recorded using a Xevo QToF G2-XS mass spectrometer (Waters UK, Manchester, UK) operated in positive ion mode. Data were processed by using MassLynx V4.1 supplied with the mass spectrometer. The relative abundance of each peptide variant was calculated as relative abundance (%) = (ion count of peptide variant/ total ion count of the five identified Aβ variants) × 100%.

### Measurements of fibril width
Twelve tomograms in-tissue fibrils with the best contrast and a Methoxy-X04 cryoCLEM signal in CLEM were used for fibril width measurements. Four tomograms from ex vivo prepared amyloid were used to measure the fibril width of sarkosyl extracted Aβ. To measure width fibrils the outside edges were manually picked in IMOD of CTF corrected tomograms. The Euclidean distance between pairs of points was computed in Matlab. Welch $t$ test ($\alpha = 0.05$) was performed using R to compare ex vivo and in-tissue tomograms. One-way ANOVA and Tukey post hoc test was performed to compare the distribution of fibril widths per tomogram.

### Reporting summary
Further information on research design is available in the Nature Portfolio Reporting Summary linked to this article.

### Data availability
The single-particle cryoEM data (refined maps, half-maps) have been deposited at the Electron Microscopy Data Bank (EMDB) under accession codes EMD-16018 (ex vivo $App^{NL-G-F}$ Aβ fibril) and EMD-16019 (control MX04-treated $App^{NL-G-F}$ Aβ fibril). The corresponding atomic models have been deposited at the Protein Data Bank (PDB) under accession codes 8BFA (ex vivo $App^{NL-G-F}$ Aβ fibril) and 8BFB (control MX04-treated $App^{NL-G-F}$ Aβ fibril). All raw cryoEM, tomographic and cryoCLEM datasets were deposited at Electron Microscopy Public Image Archive (EMPIAR) under accession codes EMPIAR-11507 (ex vivo $App^{NL-G-F}$ Aβ fibril), EMPIAR-11508 (control MX04-treated $App^{NL-G-F}$ Aβ fibril), and EMPIAR-11509 (in-tissue $App^{NL-G-F}$ β-amyloid plaque). Source data are provided with this paper.

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

## Acknowledgements

R.A.W.F. acknowledges the Academy of Medical Sciences Springboard Award (SBF005/1046), UKRI Future Leader Fellowship (MR/V022644/1) and a University of Leeds Academic Fellowship. S.E.R. holds a Royal Society Professorial Fellowship (RSRP\R1\211057). M.W. is funded by MRC (MR/T011149/1) and Y.X. by Wellcome (204963). The Astbury Biostructure Laboratory Titan Krios microscopes were funded by the University of Leeds and Wellcome Trust (108466/Z/15/Z & 221524/Z/20/Z). The Leica EM ICE, UC7 ultra/cryo-ultramicrotome and cryoCLEM systems were funded by Wellcome Trust (208395/Z/17/Z). T Xevo mass spectrometer was funded by BBSRC (BB/M012573/1). We are grateful to Takaomi Saido (RIKEN Centre for Brain Sciences) for generously providing us with the *App*^NL-G-F mouse line. We would like to thank Rebecca Thompson, Emma Hesketh, Tom O'Sullivan, Martin Fuller, Daniel Maskell, Louie Aspinall, Joshua White, Oksana Degtjarik and Yehuda Halfon for help maintaining and setting up the Astbury Biostructure Laboratory (ABSL) cryoEM facility, including cryo-preservation/imaging and Titan Krios microscopes. We are grateful to Andrew Horner, Ilona Rigo and Melanie Reay for technical support and Alex Taylor and Liam Aubrey for enlightening discussions.

## Author contributions

C.L., A.B., M.L. and R.A.W.F. organised breeding, analysed tissue, established the workflow and collected data for in-tissue cryoCLEM/cryoET. C.L., A.B., M.L., S.D. and R.A.W.F. reconstructed and analysed in-tissue cryoET data. S.G., M.W. and R.A.W.F. prepared and analysed ex vivo amyloid. M.W. collected and performed cryoEM structure determination. Y.X. performed and analysed mass spectrometry of ex vivo amyloid. M.W., and C.L. collected reconstructed ex vivo amyloid cryoET data. M.W., N.A.R and R.A.W.F. analysed ex vivo amyloid cryoET. R.A.W.F., N.A.R and S.E.R. supervised the project. All authors contributed to writing the manuscript.

## Competing interests

The authors declare no competing interests.
