## [Peer Review File · Nature Communications]

REVIEWER COMMENTS

Reviewer #1 (Remarks to the Author):

In this manuscript, the authors report the in situ structure of A β fibrillar plaques in AppNL-G-F mouse by cryo-ET, and the ex vivo structure of extracted A β fibrillar plaques by single particle reconstruction. The in situ cryo-ET structure revealed a dense network architecture of the ensembled A β polymorphs, while the ex vivo high resolution structure of the extracted A β fibrillar plaques show a different architecture from the previously reported ones, which is claimed to be caused by the mutation of E22. This study provide interesting results of amyloid plaques in the context of fresh and unfixed mammalian brain tissue, but there are some issues need to be addressed or discussed.

Major :

1. I don't quite get the point to put the in situ cryo-ET structure and the ex vivo high resolution structure together. Does the high resolution structure fit the in situ one? The relevance of these two structures, and the buffer/solution conditions difference should be commented.
2. The different ex vivo structure compared with other reported structures was claimed to be caused only by the arctic mutation (E22G) (Fig 4d-f), which is necessary to be verified by other experiments. Other possible factors, e.g., the contribution of other residues, and/or the difference in procedure/buffer conditions during extraction and structure determination process, are suggested to be indicated.
3. The A β fibrillar plaques in situ show that they are interdigitated with exosome or multilamellar bodies. This is interesting. Tomograms not in the plaque area are needed to display the difference. The relationship between the observed compartments and the A β plaque, e.g. whether A β formation could promote exosome formation, or if there is any location preference of these featured components in the plaque area than in other area, should be shown.
4. Fig. 3C, the cartoon scheme is beautifully looking. The raw tomogram superimposed with these cartoons and other data (btw, the table S1 is missing) supporting the scheme drawing need to be shown.

5. Are there any location difference of the thin protofilament-like rods and branched fibrils in situ? It would be interesting to compare the branched fibrils in this study with those amyloid fibrils with similar shapes in vitro reported by AFM, cryoEM or other techniques, such as A β , amylin, etc.

6. Fig. 4c, the extra densities around K14 (it is not labeled) are not interpreted. The residue should be labeled. Are these extra densities resulted from the fibril-fibril interaction, as reported in recent literature in different amyloid fibrils?

Minor:

1. Fig. 4b, the helical parameters including pitch and width should be labeled. More residues are necessary to be labeled in 4c-4g, particularly the E/G22 residue in Fig 4g.

2. Fig 4g, showing the interactions between protofibrils in the dimer with a magnified views of E/G22 positions may be more straightforward. The recent work (<https://doi.org/10.1101/2022.11.07.515410>) about fibrillation-promoting effects of E22G Ab could also be discussed here.

3. Extended Data Figure 2, quantitative assessment of FFT analysis in different regions are necessary.

4. Extended Data Figure 6g, the maps at 3.3 or 3.0 Å resolution look roughly identical. The densities fitted with coordinates are suggested to show to display the resolution improvement.

5. Extended Data Figure 6h, the label of local resolution map looks reversed. It does not quite fit with the quality of the density map.

6. PDB validation report should be provided.

Reviewer #2 (Remarks to the Author):

Leistner et al. investigated cryo-frozen tissue of mouse models of Alzheimer's disease using electron tomography as well as the structure of Abeta amyloid fibrils purified from these animals using single particle methods. The study is novel as there are only few such studies available. Yet, there are a number of major shortcomings and the manuscript should not be published in the current form.

The quality of the tomographic data is not always good enough to allow clear conclusions and there are many signs of artefacts introduced during sample workup. Which is the experimental evidence that the cyan features in Fig. 2b are really amyloid fibrils and not, for example, actin filaments. If these structures were actin filaments it might explain the different width values obtained with ex vivo fibrils and tomography.

Are the fibrils extracellular or intracellular? There are no clear images showing a cellular border.

The study ignores the publication by Kollmer et al. Nature Communications 2019, which was the first study of ex vivo Abeta amyloid fibrils. The study is not mentioned in the introduction, nor in the relevant results sections or in the discussion. In addition, the structural data obtained by Kollmer are not considered in the comparison of Fig. 4.

While the single particle reconstruction of the ex vivo fibrils was done well in most parts, the chirality of the fibril helix was not determined. This analysis must be carried out to be sure that the PDB model is meaningful.

Fig. 3c cannot be evaluated because the scale bar is not explained. I assume it is not the same as in panel a.

REVIEWER COMMENTS

Note: In our revised manuscript, we have changed the naming of Extended Data figures to Supplementary figures, as used in Nature Communications. We hope this is clear when referring to specific figures in our response below. Our responses are shown in blue.

Reviewer #1 (Remarks to the Author):

In this manuscript, the authors report the in situ structure of A β fibrillar plaques in AppNL-G-F mouse by cryo-ET, and the ex vivo structure of extracted A β fibrillar plaques by single particle reconstruction. The in situ cryo-ET structure revealed a dense network architecture of the ensembled A β polymorphs, while the ex vivo high resolution structure of the extracted A β fibrillar plaques show a different architecture from the previously reported ones, which is claimed to be caused by the mutation of E22. This study provide interesting results of amyloid plaques in the context of fresh and unfixed mammalian brain tissue, but there are some issues need to be addressed or discussed.

We thank the reviewer for appreciating the important results in our manuscript and for their comments and suggestions. We address each comment below and have revised the manuscript accordingly.

Major :

1. I don't quite get the point to put the in situ cryo-ET structure and the ex vivo high resolution structure together. Does the high resolution structure fit the in situ one? The relevance of these two structures, and the buffer/solution conditions difference should be commented.

The in tissue cryoET data provide the 3D molecular architecture of the β -amyloid plaque. It reveals the organisation of fibrils in situ in the context of amyloid pathology, including both amyloid- β fibrils and non-amyloid constituents, with information extending to 2-4 nm resolution. The structural properties of β -amyloid in situ were intriguing: not only did we identify classic amyloid- β fibrils, but the results also revealed the presence of protofilament-like rods and branched amyloid. The ex vivo preparation of the same tissue revealed the atomic structure of the Arctic amyloid fibril at 3.0 Å resolution. By additionally examining the same ex vivo purified sample using cryoET, we revealed rare protofilament-like rods and branched amyloid that are lost during single particle cryoEM image processing. Our results thus highlight the complementarity of the cryoEM and cryoET approaches, which enables the high-resolution interpretations to be linked with those in tissue, and allows us to interrogate the structure of amyloid at length scales from microns to sub-nanometers.

The buffer conditions are described in detail in the Methods section. In situ preparations were in a 20% dextran and an NMDG-containing artificial cerebrospinal fluid. The ex vivo amyloid used to obtain the high-resolution fibril structure was extracted with sarkosyl in a Tris.HCl-containing buffer, as used by others to extract A β fibrils previously (Yang et al, 2021).

To address the reviewer's question, and improve the clarity of our manuscript around this point, we fitted the high-resolution Arctic A β fibril structure into the in situ tomographic density of β -amyloid fibrils. The model was well accommodated considering in situ tomographic maps are of modest resolution (2-4 nm) and filaments in situ are not straight, unlike the high-resolution atomic structure which is perfectly straight because of helical symmetry averaging. We have amended the manuscript to include this point in the results and shown the atomic model fitted into the raw tomographic map in Fig. 6H of our revised manuscript. Specifically, on line 229, we now write "*The atomic structure of Arctic A β was well accommodated in the raw tomographic map of unbranched fibrils (Fig. 6h), albeit fibrils in situ showed variable degrees of curvature, whereas the ex vivo single particle Arctic A β cryoEM model has helical symmetry imposed and is thus perfectly straight (Fig. 5b)*".

2. The different ex vivo structure compared with other reported structures was claimed to be caused only by the arctic mutation (E22G) (Fig 4d-f), which is necessary to be verified by other experiments. Other possible factors, e.g., the contribution of other residues, and/or the difference in procedure/buffer conditions during extraction and structure determination process, are suggested to be indicated.

We apologise if our manuscript was not sufficiently clear on this point. A structure of ex vivo amyloid from the *App*^{NL-F} knockin mouse model of β -amyloidosis has recently been reported by Yang and coworkers (reference 13 in our manuscript). The sequence of the A β peptide from this mouse is wild-type and the structure is identical to that of human post mortem β -amyloid (type II), which also had a wild-type A β sequence. *App*^{NL-F} and the *App*^{NL-G-F} mouse strain that was investigated here are synonymous, except for the E22G mutation in the A β sequence (Saito et al., 2014; reference 16 in our manuscript). Using the same protocol, including buffers for extraction, developed by Yang et al. (ref 13 of our manuscript) for purification of *App*^{NL-F} amyloid, we purified β -amyloid from *App*^{NL-G-F} knockin mice that contain the Arctic mutation (E22G) within the A β sequence. The method of structure determination was also the same as that reported by Yang et al. (ref 13). Our experiment therefore tests directly the effect of the Arctic mutation in mouse brain, since the only difference between *App*^{NL-F} and *App*^{NL-G-F} is the E22G mutation within the A β sequence. This revealed a dramatic difference in the fibril architecture caused by the E22G mutation. It is also important to note that the Arctic E22G A β fibril structure could not accommodate the wild-type sequence because of steric clashes, as we now show in a new Supplementary Figure 10. This is consistent with the causative role of the Arctic mutation in giving rise to this novel β -amyloid structure. We have amended our revised manuscript to clarify these points.

We discuss the background of the various mutations in the Introduction (line 72):
*"These animals have a humanized mouse App gene containing three familial Alzheimer's disease mutations (Swedish, Beyruthian, and Arctic)¹⁶. The Swedish and Beyruthian mutations are located upstream and downstream of the coding region for the A β peptide, and increase the overall A β concentration, and the ratio of A β ₁₋₄₂:A β ₁₋₄₀, respectively. In contrast, the Arctic mutation is situated within the A β peptide coding sequence (App E693G, A β E22G) and is thought to increase the generation of A β protofibrils¹⁷. *App*^{NL-G-F} mice develop β -amyloid plaques, neuroinflammation, damaged synapses, and behavioural phenotypes, without ectopic over-expression of APP¹⁸."*

On Line 147, we reemphasize this:

“The structure of amyloid fibrils that result from the App^{NL-G-F} knockin strain, that only differs from App^{NL-F} in the Arctic familial AD mutation (E22G) within the Aβ₁₋₄₂ peptide, had not been reported.”

Before discussing the structural effects of the mutation on Line 169:

“In both type I and II fibrils formed from wild-type Aβ₁₋₄₂, E22 is surface exposed¹³. However, in Arctic Aβ fibrils, which only differ in the E22G mutation, G22 is buried within the fibril structure, at a point near the intra and inter-protofilament interfaces (Fig. 5d-f). Thus, it appears likely that wild-type Aβ would be unable to adopt the Arctic Aβ amyloid fold because of steric clashes created by a Glu at position 22 (Supplementary Fig. 10).”

3. The Aβ fibrillar plaques in situ show that they are interdigitated with exosome or multilamellar bodies. This is interesting. Tomograms not in the plaque area are needed to display the difference. The relationship between the observed compartments and the Aβ plaque, e.g. whether Aβ formation could promote exosome formation, or if there is any location preference of these featured components in the plaque area than in other area, should be shown.

We thank the reviewer for this helpful suggestion and, as requested, we have revised our manuscript to include a more detailed analysis of the prevalence of exosomes and multilamellar bodies in our tomographic dataset. We collected in-tissue tomograms from within and at the periphery of the methoxy-X04 cryoCLEM label of amyloid pathology in App^{NL-G-F} mice that both contained extracellular vesicles. In our revised manuscript we now also include an analysis of the prevalence of exosomes in non-AD mouse brain. This analysis showed that there were 100-fold more extracellular vesicles in App^{NL-G-F} amyloid compared to non-AD mice. In non-AD cryosections, the few extracellular vesicles that were present were exosomes and we were unable to find C-shaped membranes, ellipsoidal vesicles, multilamellar bodies, and extracellular droplets. These differences suggest that the non-amyloid constituents of amyloid plaques are indeed a feature of amyloid plaques. In the results and discussion of our revised manuscript we have included these new data reporting the higher numbers of extracellular vesicles in AD compared to non-AD tissue (new figures Fig 3c, Supplementary Fig. 4, and Supplementary Tables 1 & 2).

4. Fig. 3C, the cartoon scheme is beautifully looking. The raw tomogram superimposed with these cartoons and other data (btw, the table S1 is missing) supporting the scheme drawing need to be shown.

We apologise for the absence of Supplementary Table 1, which is now included. We have also revised and redrawn the original Fig. 3c, showing the image alone as a new Fig 4 and numbering each schematic/cartoon (i-vii). We have amended the legend to the new Fig. 4, describing the associated images, including tomographic slices corresponding to each cartoon as follows (now at line 770):

*“An extracellular cryoCLEM marker (dextran AF-647) of this amyloid plaque is shown in **Supplementary Fig. 6a**. Tomographic slices corresponding to schematics ii and vii are shown in **Fig. 3b** and **Supplementary Fig. 5a**, respectively. Tomographic slices corresponding to schematics i, iii, iv, v, and vi are shown in **Supplementary Fig. 6b-e**, respectively.”*

5. Are there any location difference of the thin protofilament-like rods and branched

fibrils in situ? It would be interesting to compare the branched fibrils in this study with those amyloid fibrils with similar shapes in vitro reported by AFM, cryoEM or other techniques, such as A β , amylin, etc.

This is an excellent, intriguing question. We did analyse whether thin fibrils (protofilament like rods) had a location difference in situ (Supplementary Fig. 12a,b in our revised manuscript). While different regions showed an enrichment in one location over another, this was not correlated with a location in core or at the periphery of methoxy-X04-labelled plaques. We looked for branched amyloid in all tomograms (19/22 tomograms) in which fibrils were oriented along the Z-axis (axial fibrils), because only in this orientation can individual fibrils be unambiguously traced through the tomogram. Branched amyloid fibrils were present in all 19 such tomograms, which included peripheral and central regions of the MX04-labelled amyloid plaques. We have included this additional detail in the results section of our revised manuscript (line 224):

“To explore whether amyloid branching exists within in-tissue tomograms, we examined the raw tomographic density of parallel fibril bundles oriented along the tomographic z-axis (19 of 22 tomograms). These provided sufficient contrast to trace individual fibrils throughout the tomographic volume (Fig. 6f), revealing branched fibrils intermingled with unbranched fibrils within amyloid plaques (Fig. 6g, h and Supplementary Fig. 13) in all 19 tomograms, regardless of their location at the periphery or in the core of a plaque.”

6. Fig. 4c, the extra densities around K14 (it is not labelled) are not interpreted. The residue should be labelled. Are these extra densities resulted from the fibril-fibril interaction, as reported in recent literature in different amyloid fibrils?

We apologise for the oversight of not labelling these residues in Fig. 4c (now Fig. 5), we have amended our revised manuscript accordingly.

The unknown peripheral densities are highly unlikely to be related to fibril-fibril interactions because the vast majority of particles contributing to the final structure were picked from locations containing a single fibril, no evidence of fibril-fibril bundling was evident, and such density would perforce be weaker than the fibril itself.

As we are sure the reviewer is aware, many atomic structures of amyloid from mouse and human tissue reveal the presence of unknown density (eg Fitzpatrick et al., Nature, 2017 and Yang et al., Science, 2022). In common with this literature precedent (and many others), we cannot say with any certainty what the additional density represents other than it is likely a negatively charged molecule and is larger than the small ions seen on the surface of A β ₄₂ fibrils extracted from human patients in the study by Yang et al., 2022. Therefore, we have avoided speculation on the subject in our manuscript.

Minor:

1. Fig. 4b, the helical parameters including pitch and width should be labeled. More residues are necessary to be labeled in 4c-4g, particularly the E/G22 residue in Fig 4g.

Thank you for the suggestion, we have included more residue labelling as requested.

2. Fig 4g, showing the interactions between protofibrils in the dimer with a magnified

views of E/G22 positions may be more straightforward. The recent work (<https://doi.org/10.1101/2022.11.07.515410>) about fibrillation-promoting effects of E22G Ab could also be discussed here.

We have expanded Fig. 4, (now Fig. 5) to include the ex vivo Abeta1-40 model (Kollmer et al., 2019) and modified it by colour-matching a chain of each dimeric fibril model to those in the superposition (now Fig. 5h) in addition to highlighting residue position 22. We believe the changes help to make the figure more informative with Fig. 5d-5g giving a view of the global context of the positioning of residue 22 in each model and Fig. 5h showing the region of similarity within the subunit folds. We are happy to look at the figure again with regards to showing magnified views of the E/G22 residue if required, but it is not our feeling that this is now needed. As requested, we have also included a brief discussion of the recently published work mentioned by the reviewer that is consistent with our manuscript in line 253 of our revised manuscript: *“The atomic model is similar to a recent report of the same amyloid³⁴ and showed that App^{NL-G-F} fibrils of the Arctic variant have a different fold to that observed in the App^{NL-F} knockin model, indicating that the Arctic FAD mutation (E22G) causes a dramatic change in the fibril structure.”*

3. Extended Data Figure 2, quantitative assessment of FFT analysis in different regions are necessary.

We have included this in a new panel (Supplementary Fig. 2c) of our revised manuscript as requested. We find that the distance between fibrils within parallel bundles is variable from one region the amyloid plaque to the next.

4. Extended Data Figure 6g, the maps at 3.3 or 3.0 Å resolution look roughly identical. The densities fitted with coordinates are suggested to show to display the resolution improvement.

The reviewer is correct that the 0.3 Å resolution improvement is not obvious in the map. In fact we reviewed the processing steps for both maps and found we had mistakenly used a subtly different value in postprocessing/sharpening between the two jobs. We apologise for this error and we have repeated the steps with the correct sharpening parameter, and updated a new Supplementary figure (now Supplementary Fig. 7). The difference in resolution is now 3.1 vs 3.0 Å and makes no substantive difference to the result. The two maps displayed the same sharpening level are now almost indistinguishable.

5. Extended Data Figure 6h, the label of local resolution map looks reversed. It does not quite fit with the quality of the density map.

The reviewer is correct and we thank them for spotting our error. We have corrected this error in the revised manuscript (note this figure is now Supplementary Fig. 7).

6. PDB validation report should be provided.

We apologise for the delay in providing these. They are included with the revised manuscript.

Reviewer #2 (Remarks to the Author):

Leistner et al. investigated cryo-frozen tissue of mouse models of Alzheimer's disease using electron tomography as well as the structure of Abeta amyloid fibrils purified from these animals using single particle methods. The study is novel as there are only few such studies available. Yet, there are a number of major shortcomings and the manuscript should not be published in the current form.

We thank the reviewer for recognising the novelty of our manuscript in that we report the first in-tissue cryoET study of amyloid in the mammalian brain. We also thank the reviewer for their comments and questions, which we have addressed below.

The quality of the tomographic data is not always good enough to allow clear conclusions and there are many signs of artefacts introduced during sample workup. Which is the experimental evidence that the cyan features in Fig. 2b are really amyloid fibrils and not, for example, actin filaments. If these structures were actin filaments it might explain the different width values obtained with ex vivo fibrils and tomography.

The reviewer asks an important question.

Several lines of evidence give us confidence in our conclusions. In-tissue/cryo-section tomograms were all collected using methoxy-X04 as a cryoCLEM label (Supplementary Table 1). We tested this well-characterised amyloid label showing that methoxy-X04 labelled deposits were absent in all tissue samples of wild-type mice not expressing APP in our cryoCLEM work-flow (Fig. 1c and Supplementary Fig. 1c). If methoxyX04 labelled non-amyloid proteins, including actin, these would give a fluorescent signal in the wild-type samples, which we did not observe (Fig. 1c and Supplementary Fig. 1c). This indicates that methoxy-X04 cryoCLEM labels only amyloid. We also confirmed by immunofluorescence that the plaque pathology methoxy-X04 labels in the App^{NL-G-F} model is composed of A β (Supplementary Fig. 1a-b).

Specifically, the tomogram shown in Fig 2b was collected from the methoxy-X04 cryoCLEM label shown in Figure 1c (see yellow box showing the location from which the tomographic tilt series was collected).

Finally, we have collected tomograms from forebrain of control mice that do not develop amyloid plaques (Peukes et al., bioRxiv (<https://doi.org/10.1101/2021.02.19.432002>)). We have now supplemented this dataset with a total of 40 tomograms that we have now analysed. Extracellular filamentous structures were not observed in any of these data. We have included an analysis of these data in our revised manuscript. In results section (line 120):

“We next quantified whether extracellular vesicles were more common in MX04-labelled amyloid plaques than in non-AD tissue. We previously reported the in-tissue molecular architecture of mouse brain by cryoET of cryo-sections from *Psd95^{GFP}* knockin mice that do not contain FAD mutations (*App^{WT/WT}*)²⁷. This mouse strain labels glutamatergic synapses with GFP-tagged PSD95 but are otherwise without fibrils, amyloid plaques, or any other abnormal phenotype. Click or tap here to enter text.²⁷⁻²⁹. Comparing the prevalence of extracellular vesicles in *App^{NL-G-F}* in-tissue tomograms with those from *App^{WT/WT} - Psd95^{GFP/GFP}* mice (Supplementary Tables 1 & 2) indicated that there were on average 100-fold more

extracellular vesicles in MX04-labelled amyloid plaques compared to that of tissues lacking β -amyloid plaque pathology ($P < 0.005$, mean=10.6 and 0.1, $n=18$ and $n=40$ in App^{NL-G-F} and 40 $App^{WT/WT} - Psd95^{GFP}$ in-tissue tomograms, respectively, **Fig. 3c** and **Supplementary Fig. 4**). Interestingly, only exosomes were present in $App^{WT/WT} - Psd95^{GFP}$, whereas C-shaped membranes, ellipsoidal vesicles, multilamellar bodies, and extracellular droplets were absent in tissues lacking amyloid plaques (**Supplementary Fig. 4**)²⁶.”

And in the Methods section, line 449:

“The workflow for collecting in-tissue tomograms from $App^{WT/WT} - Psd95^{GFP/GFP}$ mice forebrain (cortex and hippocampus) cryo-sections was as previously described²⁷ and was the same as that described for App^{NL-G-F} mice, but without the i.p. administration of MX04. To quantify extracellular vesicles in tomograms from $App^{WT/WT} - Psd95^{GFP/GFP}$ mice (**Supplementary Table 2**), the number of exosomes, extracellular vesicles, ellipsoidal vesicles, multilamellar bodies and extracellular droplets were counted and compared to those from App^{NL-G-F} mice.”

Are the fibrils extracellular or intracellular? There are no clear images showing a cellular border.

This is also an important question. We apologise that the discussion of this feature of our data, and the conclusions we draw from it, was not sufficiently clear. In our cryoET data the lipid bilayer of membranes was unambiguous, except when that membrane was oriented on or near to the x-y plane of the tomogram. Plasma membranes were distinguishable from other membranes (e.g. exosomes) since they enclosed a compartment containing ribosomes, microtubules, mitochondria or rough endoplasmic reticulum. These consistent criteria in tracing plasma membranes, ribosomes and organelles revealed that all of the $A\beta$ fibrils were in extracellular locations and certainly not in the cytoplasm of cells in any of our tomograms. To clarify this point, we have amended the Methods section in our revised manuscript (line 436):

“Cell plasma membranes: marked the boundary between the higher tomographic density of the cytoplasm and the lower tomographic density of extracellular interstitial space, as well as enclosing ribosomes, microtubules, mitochondria, and rough endoplasmic reticulum.”

In some tomograms of the core of methoxy-X04-labelled amyloid plaque, no plasma membrane-bound cells were evident. Therefore, to verify that the amyloid we detected in our methoxy X04 cryoCLEM workflow corresponded to extracellular amyloid plaques in App^{NL-G-F} brain samples we used an additional cryoCLEM label (dextran AF647) to label the extracellular space in our tissue (Fig. 1d and Supplementary Fig. 6a in our revised manuscript). Dextran AF647 is a fluorescently labelled polymer that cannot cross plasma membranes (Nicholson and Tao, 1993 (doi: 10.1016/S0006-3495(93)81324-9)), and its inclusion indicated that all of the methoxy-X04 stained amyloid in our tomograms is extracellular.

To clarify this point, we have moved our cryoCLEM data using dextran AF647 as an extracellular marker from the Supplementary figures into Fig. 1d of our revised manuscript, so that this important information is more prominent. We have also amended our results section on line 91:

“To verify that App^{NL-G-F} amyloid formed as extracellular deposits, we labelled tissue using an extracellular fluorescent marker, dextran-AF647 that is incapable of crossing the plasma membrane²³. CryoEM of tissue cryo-sections from these samples indicated MX04 overlapping with dextran-AF647, confirming the extracellular location of amyloid (Fig. 1d).”

The study ignores the publication by Kollmer et al. Nature Communications 2019, which was the first study of ex vivo Abeta amyloid fibrils. The study is not mentioned in the introduction, nor in the relevant results sections or in the discussion. In addition, the structural data obtained by Kollmer are not considered in the comparison of Fig. 4.

We agree both that Kollmer et al. was the first study to report the structure ex vivo Abeta amyloid fibrils, and of the critical relevance of this work to the results we present. This was an error on our part for which we apologise. In fact we incorrectly cited an article in Nature Communications that accompanied the Kollmer paper (Liberta et al., 2019) from the same group and we thank the reviewer for spotting our error. We have included the correct citation in our revised manuscript.

In preparing our original manuscript, we did perform an alignment of ex vivo CAA amyloid (Abeta1-40) with the available Abeta1-42 structures, but these aligned poorly, suggesting that the Abeta1-40 structure was structurally very different from that of Abeta1-42. In our revised manuscript we have included a comparison of all ex vivo structures of Abeta, including Abeta1-40, as requested (revised Fig. 5h).

While the single particle reconstruction of the ex vivo fibrils was done well in most parts, the chirality of the fibril helix was not determined. This analysis must be carried out to be sure that the PDB model is meaningful.

We have now determined the hand of the fibril twist by collecting cryoET data of ex vivo Arctic amyloid (the same sample used for our single particle reconstruction). The results confirm that the Arctic A β fibrils have a left-handed twist, as is seen for A β 1-42 fibrils from App^{NL-F} mice and post mortem AD brain (reference 13 in our manuscript). However, it differs from the right-handed twist observed in A β 1-40 fibrils from post mortem CAA meninges reported by Kollmer et al. (discussed above). We have included this new information in our revised manuscript on line 163:

*“App^{NL-G-F} ex vivo fibrils, which had a left-handed twist (as determined by cryoET, **Supplementary Fig. 9 and Methods**)”*

And a new sentence in the Methods on line 517:

*“The hand of the App^{NL-G-F} ex vivo purified amyloid fibril twist was determined by cryoET (**Supplementary Fig. 9**). The final model quality was assessed using MolProbity⁶⁴ and the results detailed in **Supplementary Table 3**.”*

Fig. 3c cannot be evaluated because the scale bar is not explained. I assume it is not the same as in panel a.

We apologise for omitting a description of the scale bar in Fig. 3c. We have included this information in the figure legend of our revised manuscript (now Fig. 4).

REVIEWER COMMENTS

Reviewer #1 (Remarks to the Author):

I thank the authors for the well prepared responses and the supplied additional details. Most of my issues have been addressed and I am overall satisfactory with it.

The remaining points I have are:

1) The relevance between in vivo and ex vivo results. I appreciate the efforts of in vivo cryo-ET studies by the authors and understand it's not easy to achieve a higher resolution tomograms. However, since the relevance between in vivo and ex vivo reconstructions is one of the key points of the manuscript, showing a picture of tomogram of fibers fitted with the high resolution helical structure is necessary. If possible, some further data processing of the cryoET data, e.g. sub-tomo classification and average of different types of amyloid fibril, would be more favorable.

2) The accuracy of plaque compartments annotation in cryoET data. It is understandable that the current cryoET data could not provide very detailed structural features owing to their relatively low resolution, so the description or statement could be a little toned down, too.

Minor:

The resolutions before and after CTF refinement and Bayesian polishing are now 3.1 Å vs 3.0 Å as mentioned in the responses to my previous minor 4, while they still show as 3.3 Å and 3.0 Å in the revised Supplementary Fig. 7g respectively,

Reviewer #2 (Remarks to the Author):

The authors addressed all points and I have no further comments.

NCOMMS-22-45304A

The in-tissue molecular architecture of β -amyloid pathology in the mammalian brain
Response to reviewers

Reviewer #1.

I thank the authors for the well prepared responses and the supplied additional details. Most of my issues have been addressed and I am overall satisfactory with it.

The remaining points I have are:

1) The relevance between *in vivo* and *ex vivo* results. I appreciate the efforts of *in vivo* cryo-ET studies by the authors and understand it's not easy to achieve a higher resolution tomograms. However, since the relevance between *in vivo* and *ex vivo* reconstructions is one of the key points of the manuscript, showing a picture of tomogram of fibers fitted with the high resolution helical structure is necessary.

The referee requests a figure with our *ex vivo* high-resolution structure determined by cryoEM, fitted into the *in situ* tomograms. This was in fact included in Fig. 6h, but this was obviously not sufficiently clear, and we apologise for this. We have completely remade this figure, including the near-atomic structure fitted into several examples of fibrils in tomograms in a new Supplementary Fig. 12c. We hope that this shows the excellent accord between the unbranched fibrils *in situ* and atomic model from *ex vivo* preparations of amyloid from the same source, and that this will be much clearer to the reader.

If possible, some further data processing of the cryoET data, e.g. sub-tomo classification and average of different types of amyloid fibril, would be more favorable.

The reviewer recognizes the difficulty of achieving higher resolution tomograms, and we agree. The tomograms presented represent the current state of the art in this area and there is no immediate prospect of improvement in the raw data.

The idea of sub-tomogram averaging is an excellent one but based on extensive tests that we have performed with this and other data, **we do not believe it is technically possible for amyloid fibrils at present**. Sub-tomogram averaging of molecules in cultured mammalian cells has only achieved results better than ~ 20 Å for ribosomes. Ribosomes are large, full of electron-dense RNA (which enhances contrast), and have protein and RNA domains that provide structural features of various sizes that aid alignment and averaging. No report of anything approaching this resolution has been made for frozen sections, which are necessarily thicker. For amyloid the problem is more challenging even in cells, but are prodigious in tissue cryo-sections. In particular, amyloid fibrils are rather featureless, with the only significant structural features being the gross helical twist – a cross-over distance of ~ 660 Å in this case, and the 4.8 Å β -spacing of amyloid layers. Given that we estimate the maximum resolution of the tomograms at around 20 Å, in our hands sub-tomogram averages generating ~ 30 Å structures, but given the nature of amyloid structures, these are featureless tubes that convey no new information. There is no literature precedent for sub-tomogram averaging *in situ* amyloid.

2) The accuracy of plaque compartments annotation in cryoET data. It is understandable that

the current cryoET data could not provide very detailed structural features owing to their relatively low resolution, so the description or statement could be a little toned down, too.

We agree. We have made several changes to the text to make claims less definitive:

In Results section, Line 135: "In these tomographic volumes the boundary between intra- and extracellular region of tissue was marked by plasma membranes that enclosed characteristic cytoplasmic constituents that were readily identifiable (see Methods), including putative ribosomes, microtubules, as well as rough endoplasmic reticulum and mitochondria (Figs. 3 & 4, Supplementary Figs. 5 & 6)."

Methods section, line 430:

"Putative ribosomes were distinguishable as ~30 nm particles with higher tomographic density than surrounding macromolecules, in accordance with their high nucleic acid content."

Line 434: "Subcellular compartments were defined by containing a higher tomographic density than the extracellular space that is consistent with characteristic of the higher concentration of proteins in the cell cytoplasm compared to the extracellular space."

Line 437: "Cell plasma membranes: marked the boundary between the higher tomographic density of the cytoplasm and the lower tomographic density of extracellular interstitial space, as well as enclosing putative ribosomes, microtubules, mitochondria, and rough endoplasmic reticulum."

Line 443: "The extracellular versus intracellular topology of locations within cryoET data was established on the basis of macromolecular and organelle constituents that was consistent with the expected are characteristic of cell cytoplasmic, including microtubules, putative ribosomes, rough endoplasmic reticulum, and mitochondria."

Minor:

The resolutions before and after CTF refinement and Bayesian polishing are now 3.1 Å vs 3.0 Å as mentioned in the responses to my previous minor 4, while they still show as 3.3 Å and 3.0 Å in the revised Supplementary Fig. 7g respectively,

Yes, we apologise for this. We have corrected this error on our part with a properly revised figure.